# InfoPO: Information-Driven Policy Optimization for User-Centric Agents

**Fanqi Kong** [1][2] **Jiayi Zhang** [2][3] **Mingyi Deng** [2] **Chenglin Wu** [2] **Yuyu Luo** [3] **Bang Liu** [4]

## Abstract

Real-world user requests to LLM agents are often underspecified. Agents must interact to acquire missing information and make correct downstream decisions. However, current multi-turn GRPO-based methods often rely on trajectory-level reward computation, which leads to credit assignment problems and insufficient advantage signals within rollout groups. A feasible approach is to identify valuable interaction turns at a fine granularity to drive more targeted learning. To address this, we introduce InfoPO (Information-Driven Policy Optimization), which frames multi-turn interaction as a process of active uncertainty reduction and computes an information-gain reward that credits turns whose feedback measurably changes the agent's subsequent action distribution compared to a masked-feedback counterfactual. It then combines this signal with task outcomes via an adaptive variance-gated fusion to identify information importance while maintaining task-oriented goal direction. Across diverse tasks, including intent clarification, collaborative coding, and tool-augmented decision making, InfoPO consistently outperforms prompting and multi-turn RL baselines, exceeding GRPO-based methods by 14% to 16%. It also demonstrates robustness under user simulator shifts and generalizes effectively to environment interactive tasks. Overall, InfoPO provides a principled and scalable mechanism for optimizing complex agent user collaboration. Code is available at https://github.com/kfq20/InfoPO.

## 1. Introduction

The rapid evolution of Large Language Models (LLMs) has enabled interactive agents that assist users in complex,

[1]Peking University [2]DeepWisdom [3]The Hong Kong University of Science and Technology (Guangzhou) [4]Université de Montréal & Mila. Correspondence to: Bang Liu <bang.liu@umontreal.ca>.

*Proceedings of the 43rd International Conference on Machine Learning*, Seoul, South Korea. PMLR 306, 2026. Copyright 2026 by the author(s).

multi-turn tasks (Zhu et al., 2025; Li et al., 2025a; Liu et al., 2025a; Li et al., 2025e). In the user-centric setting where agents must serve human users with underspecified goals, they must bridge a fundamental gap between often ambiguous human intentions and the precise parameters required for machine execution (Norman, 1986; Clark & Brennan, 1991; Budzianowski et al., 2018). For example, a request like "book me a flight next week" is not directly actionable until the agent elicits missing constraints such as dates, departure airport, budget, and flexibility. Therefore, an effective interaction should both increase the agent's knowledge about the user's true intent and advance the task toward completion (Kong et al., 2025). Mastering this interplay between intent elicitation and task execution remains a core challenge for building autonomous agents that operate reliably under partial information (Deng et al., 2025; Qian et al., 2025b; He et al., 2025; Luo et al., 2018).

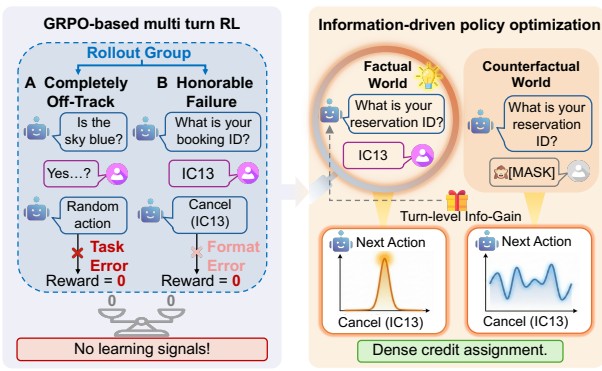

*Figure 1.* Standard GRPO vs. InfoPO. Standard GRPO yields zero reward for "honorable failures" (correct elicitation, failed execution). InfoPO solves this via counterfactual masking to provide dense, turn-level information-gain rewards.

This challenge calls for a principled approach to learning an interaction policy. Reinforcement learning (RL) stands out as the standard paradigm for this purpose, as it enables an agent to autonomously discover effective strategies from feedback in sequential decision-making settings (Kong et al., 2024; Zhang et al., 2025a; Liu et al., 2025b). However, applying RL to multi-turn agentic tasks exposes its Achilles' heel: the long-horizon credit assignment problem. Rewards are often sparse and delayed until task completion, making it difficult to attribute outcomes to intermediate decisions (Zhou et al., 2025; Wei et al., 2025). This issue is

particularly pronounced under GRPO-based methods (Shao et al., 2024; Yu et al., 2025), where policy updates rely on reward variation within a rollout group. Moreover, many recent works on multi-turn RL aggregate terminal and intermediate signals into a single trajectory-level score for advantage estimation, which limits fine-grained supervision across the interaction (Qian et al., 2025a;c; Wang et al., 2025; Jin et al., 2025; Lin et al., 2025). In user-centric environments, such granularity matters even more. A small number of clarification decisions can determine feasibility and downstream success, and RL training frequently relies on LLM-simulated users (Zhao et al., 2025a; Cai et al., 2025; Li et al., 2025d;c), making sample efficiency also a critical constraint.

To address the above limitation, we propose Information-Driven Policy Optimization (**InfoPO**), which treats multi-turn interaction as a process of active uncertainty reduction (as illustrated in Figure 1). InfoPO defines a turn-level *counterfactual information-gain* reward that credits the action for the information it elicits. After the agent acts, the user or environment provides feedback. We then ask how this feedback would change the agent's next decision. Concretely, we score the same collected trajectory under two conditions: (i) the *factual history* that includes the feedback and (ii) a *counterfactual history* where the feedback is replaced by a masked placeholder. We compare the policy's probability assigned to the actual next action under the two conditions. The resulting distribution gap is attributed to the action that triggered the feedback, rewarding turns that meaningfully reshapes downstream choices. This process does not need additional interactions with environments.

To keep this intrinsic signal aligned with task completion, InfoPO uses an adaptive *variance-gated fusion* to combine information gain with outcome-based updates. When outcome rewards within a rollout group are non-discriminative, group-relative advantages can be near zero, leading to learning stagnation. In this case, the gate increases the weight on information gain to maintain a usable training signal. As outcomes become more discriminative, the gate shifts weight back to the task objective to support eventual success.

We also provide an information-theoretic interpretation of the learning signal. In expectation, the per-turn information gain reward corresponds to a conditional mutual information between feedback and the agent's next action, while its cumulative form represents the directed information flow from observations to decisions. Crucially, we prove that a minimum cumulative information gain is a strictly necessary resource for achieving task success, providing a theoretical lower bound that links uncertainty reduction directly to the probability of reaching the goal.

We evaluate InfoPO on three representative interactive benchmarks, UserGym (Qian et al., 2025c), Col-

Bench (Zhou et al., 2025), and the long-horizon $\tau^2$-Bench (Barres et al., 2025), which together span intent clarification and preference elicitation, collaborative code generation, and tool-augmented decision making. InfoPO consistently improves task performance and learning stability over strong prompting and RL baselines. To attribute these gains, we run component ablations and diagnostic analyses, uncovering a learned interaction pattern that resolves intent uncertainty early before downstream commitments. We further test robustness beyond user-centric benchmarks by applying InfoPO to non-user-interactive tasks and by swapping the simulated user at evaluation time, indicating that InfoPO learns broadly useful interaction strategies.

In summary, our contributions are threefold: **(1)** we introduce InfoPO, an information-driven RL method for multi-turn interaction that provides dense turn-level credit to enable effective learning under sparse or delayed outcomes; **(2)** we provide an information-theoretic grounding that links the proposed signal to conditional mutual information and directed information, clarifying the role of information flow from feedback to actions in learning; **(3)** we deliver a comprehensive empirical study with ablations and diagnostic analyzes across multiple interactive benchmarks, showing improved task performance and training stability, and generalization to variant users and environments.

## 2. Related Works

**User-centric agents.** User-centric agents move beyond "task completion" toward inferring latent intent, preferences, and user state under multi-turn interaction. On the intent/need side, recent work studies implicit intention elicitation and clarification policies for agents (Qian et al., 2024; Chen et al., 2024; Li et al., 2025b; 2024), and connects user-facing transparency/explanations to predictability and controllability in personalized settings (Yu et al., 2024; Hong & Roth, 2026; Qin et al., 2020). On the personalization side, the community is rapidly enriching benchmarks and problem formulations, from classic persona-grounded dialogue (Zhang et al., 2018) to modern LLM-centric personalization suites such as LaMP/LaMP-QA (Salemi et al., 2024; Salemi & Zamani, 2025), and preference-heterogeneity benchmarks for individualized alignment (Zollo et al., 2024; Afzoon et al., 2024; Wu et al., 2024). In realistic user assistance, goals are embedded in open-ended workflows that require iterative feedback incorporation, grounded actions, and goal adaptation under practical constraints (e.g., interacting with external information, code, services, or APIs), motivating multi-turn testbeds that jointly evaluate these capabilities (Barres et al., 2025; Qian et al., 2025b;c; Wu et al., 2025; He et al., 2025; Xie et al., 2025). A complementary line builds agent training/evaluation scaffolds via user simulators, long-term memory, and synthetic environment generation (Sun et al.,

2025; Li et al., 2025d; Cai et al., 2025; Wu et al., 2026; Zhang et al., 2025b). These advances highlight both the promise of group-relative RL and the open challenge of fine-grained credit assignment in long, interactive rollouts.

**Agentic reinforcement learning.** Reinforcement learning has become a broadly applicable approach to improve LLM agents' decision-making across diverse tasks. Multi-turn credit assignment is studied via hierarchical and collaborative training (Zhou et al., 2024; 2025), and grounded tool use and information seeking are induced with RL across search, clarification, and tool actions (Jin et al., 2025; Zhao et al., 2025a; Acikgoz et al., 2025; Ruan et al., 2026); analyses of long-rollout dynamics further identify instability and motivate stabilization (Wang et al., 2025). Optimization has shifted toward RLVR-style designs for long, high-variance trajectories: group-relative methods and refinements (Shao et al., 2024; Feng et al., 2025), sequence/length-normalized stabilization (Zheng et al., 2025; Zhao et al., 2025b), training efficiency (Yu et al., 2025; Sheng et al., 2025), and variants relaxing strict group synchronization (Xu & Ding, 2025). For tool interaction, ARPO/AEPO handle post-tool uncertainty with entropy-aware rollout and update stabilization (Dong et al., 2025a;b); for multi-reward settings, GDPO decouples normalization (Liu et al., 2026) and ARIA aggregates rewards by intention (Yang et al., 2025b). InfoPO proposes a turn-level information advantage for credit assignment in user-centric multi-turn interaction, avoiding task-specific dense shaping or process reward models.

**Reward Shaping in RL.** Reward shaping is a common way to speed up learning when rewards are sparse. A classic approach adds intrinsic signals to promote information seeking, such as curiosity bonuses based on novelty or prediction error (Pathak et al., 2017; Burda et al., 2018), or empowerment-style objectives that increase future controllability (Klyubin et al., 2005; Mohamed & Jimenez Rezende, 2015). In LLM post-training, similar ideas appear as preference-based supervision, which is denser than end-task success alone; process reward models (PRMs) push this further to the step level by scoring intermediate reasoning steps (Ma et al., 2023; Khalifa et al., 2025; Xi et al., 2025). Step-wise feedback has shown clear gains on complex reasoning (e.g., math), improving verification and self-correction by reducing compounding errors (Setlur et al., 2024; Ye et al., 2025). InfoPO derives a turn-level learning signal without requiring task-specific heuristics.

## 3. Preliminaries

### 3.1. Multi-Turn Interaction as a Dec-POMDP

We model the *user-centric* multi-turn task as a decentralized partially-observable Markov decision process (Dec-POMDP) (Bernstein et al., 2002). In this setting, the agent repeatedly interacts with a user (or simulator) to infer latent intent $Z$. At each turn $t$, the environment transitions to a latent state $s_{t+1} \sim \mathcal{P}(\cdot \mid s_t, a_t)$ and reveals an observation $o_t \sim \mathcal{O}(\cdot \mid s_{t+1})$. Let $h_t = (q, a_1, o_1, \ldots, a_{t-1}, o_{t-1})$ denote the interaction history before observing $o_t$. The policy $\pi_\theta(a_t \mid h_t, o_t)$ generates a sequence of tokens conditioned on the transcript. Crucially, in user-centric tasks, the observation $o_t$ reduces uncertainty about the task goal and shifts the action distribution for the next turn $a_{t+1}$.

### 3.2. Optimization with Group-Relative Policy Gradient

The objective is to maximize the expected external return $J(\theta) = \mathbb{E}_{\tau \sim \pi_\theta}[R^{\text{ext}}(\tau)]$, where $R^{\text{ext}}(\tau)$ is the cumulative reward. For LLM agents, we update $\theta$ at the token level using an advantage signal $A_{i,k}$:

$$\nabla_\theta \mathcal{L}(\theta) \propto \mathbb{E}\left[\sum_{k=1}^{L_i} A_{i,k} \nabla_\theta \log \pi_\theta(y_{i,k} \mid x_i, y_{i,<k})\right]. \quad (1)$$

InfoPO is built upon Group-Relative Policy Optimization (GRPO) (Shao et al., 2024), which computes $A_{i,k}$ by comparing trajectories within a rollout group to estimate variance-reduced advantages without an explicit critic.

## 4. Methods

We propose **InfoPO**, a multi-turn RL algorithm for user-centric interaction where intent and constraints are revealed through feedback. As shown in Figure 2, InfoPO introduces a **turn-level counterfactual info-gain reward** that credits actions by how much the received observation changes the policy's next-step decision under a masked-observation counterfactual. InfoPO further applies a **variance-gated fusion** strategy that adaptively combines info-gain and outcome-based learning based on the within-group discriminativeness of external rewards. Algorithm 1 in the appendix summarizes the full procedure.

Section 4.1 defines the counterfactual info-gain reward. Section 4.2 presents advantage estimation and the variance-gated fusion that yields the final objective. Section 4.3 provides an information-theoretic interpretation and establishes the necessity of information gain for task success.

### 4.1. Turn-level Counterfactual Information Gain

A key challenge is to define a task-agnostic measure of information progress. We posit that a high-quality observation should *reduce the uncertainty* of the agent's task state, which is reflected in the shift of its subsequent action distribution. By comparing the real transcript to a counterfactual one where the observation is absent, we can isolate the specific "information gain" attributed to that turn.

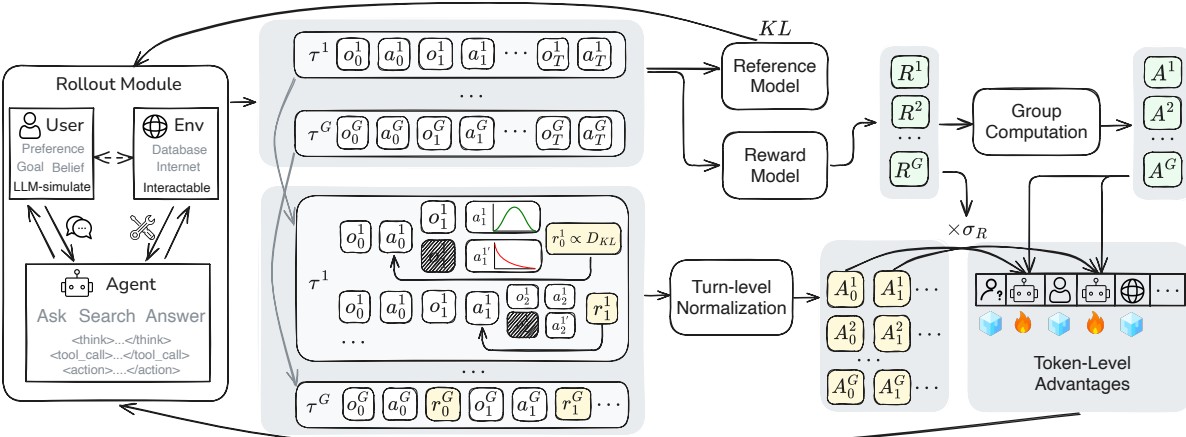

*Figure 2.* Overview of the InfoPO framework. It extracts a turn-level info-gain signal by counterfactual reasoning and adaptively fuses it with outcome-based advantages to facilitate efficient credit assignment in multi-turn user-centric tasks.

At turn $t$, the agent produces an action segment $a_t$ and receives feedback $o_t$. Let $a_{t+1} = (y_1, \ldots, y_{L_{t+1}})$ be the realized next action token sequence. We define the **turn-level info-gain reward** $r_t^{\text{info}}$ as the average log-probability shift over the next action tokens:

$$r_t^{\text{info}} = \frac{1}{L_{t+1}} \sum_{k=1}^{L_{t+1}} \Big( \log \pi_\theta(y_k \mid h_t, o_t, y_{<k}) \tag{2}$$
$$- \log \pi_\theta(y_k \mid h_t, \varnothing, y_{<k}) \Big),$$

where $\varnothing$ is a string placeholder named "No information found." Statistical analysis in Appendix B.3 confirms that our method is robust to different placeholder implementations. Both terms in Eq. 2 are computed using *teacher forcing* on the same realized tokens $a_{t+1}$. This design serves two purposes: (1) *Causal Isolation*: it ensures that the shift in $r_t^{\text{info}}$ is strictly caused by the presence of $o_t$ rather than stochastic variations in autoregressive generation; (2) *Computational Tractability*: by using the same tokens, we avoid the prohibitive cost of multiple autoregressive rollouts for each counterfactual turn, allowing $r_t^{\text{info}}$ to be computed efficiently via parallelized forward passes.

InfoPO does not require an external oracle to pre-identify whether an action is meaningful. The information-gain reward is computed for every realized transition by comparing the policy's likelihood of the same next action under factual and masked-feedback contexts. However, as with standard trial-and-error RL, the method assumes that the behavior policy and the user/environment have non-zero support over task-relevant continuations. If all sampled trajectories are completely degenerate or all feedback is uninformative, no intrinsic reward can recover the missing task structure. In practice, current instruction-tuned LLM agents usually produce a mixture of useful, failed-but-grounded, and meaningless actions, and we empirically verify in Appendix B.5

that high information gain is concentrated on task-relevant transitions rather than arbitrary distribution shifts.

### 4.2. Unified Group-Relative Advantage Construction

InfoPO uses group-relative estimation to stabilize updates without a learned critic. For each prompt, we sample a group of $G$ trajectories $\tau_1, \ldots, \tau_G$ and form advantages by comparing trajectories within the same context. We then compute the outcome-based advantage and the info-gain advantage separately:

**Outcome Advantage.** Let $R_i^{\text{ext}}$ be the trajectory-level external reward for rollout $i$. We compute the normalized outcome advantage $A_{i,k}^{\text{ext}}$ for each token $k$ in the response:

$$A_{i,k}^{\text{ext}} = \frac{R_i^{\text{ext}} - \mu_g^{\text{ext}}}{\sigma_g^{\text{ext}} + \epsilon} \cdot m_{i,k}, \tag{3}$$

where $\mu_g^{\text{ext}}$ and $\sigma_g^{\text{ext}}$ are the mean and standard deviation of external scores within group $g$, $m_{i,k}$ is a response mask, and $\epsilon$ is a small constant to avoid division by zero.

**Info-Gain Advantage.** We normalize turn-level info-gain rewards within group $g$ and broadcast the resulting scalar to the tokens of the corresponding action segment:

$$A_{i,k}^{\text{info}} = \frac{r_{i,t(k)}^{\text{info}} - \mu_g^{\text{info}}}{\sigma_g^{\text{info}} + \epsilon} \cdot m_{i,k}, \tag{4}$$

where $t(k)$ maps token $k$ to its associated interaction turn, and $(\mu_g^{\text{info}}, \sigma_g^{\text{info}})$ are computed over all valid turns in group.

**Adaptive Fusion via Variance Gating.** We then combine the two advantages into a single update signal. InfoPO uses an adaptive gate $g(\cdot)$ to scale the info-gain contribution

according to the within-group variability of the outcome signal: $g(\sigma_g^{\text{ext}}) = \sigma\left(-\frac{\sigma_g^{\text{ext}}}{T}\right)$, where $T$ is a temperature parameter. When the external outcomes within a group are nearly indistinguishable ($\sigma_g^{\text{ext}} \approx 0$), $g(\cdot)$ increases, allowing the info-gain advantage to drive learning. Conversely, as the outcome signal becomes more discriminative, $g(\cdot)$ approaches 0, and the policy prioritizes task success. The final unified advantage for token $k$ in rollout $i$ is:

$$\hat{A}_{i,k} = A_{i,k}^{\text{ext}} + \beta \cdot g(\sigma_g^{\text{ext}}) \cdot A_{i,k}^{\text{info}}, \qquad (5)$$

where $\beta$ controls the peak influence of the information progress. This design encourages informative interaction while keeping optimization anchored to the task objective.

Finally, we optimize the policy $\pi_\theta$ relative to the reference policy $\pi_{\text{ref}}$ to control distribution shift. The unified objective of InfoPO is formulated as:

$$\mathcal{J}_{\text{InfoPO}}(\theta) = \mathbb{E}_{q \sim P(Q), \{\tau_i\}_{i=1}^G \sim \pi_{\theta_{\text{old}}}} \left[ \frac{1}{G} \sum_{i=1}^{G} \frac{1}{|\tau_i|} \sum_{k=1}^{|\tau_i|} \Bigg\{$$

$$\min\left( \frac{\pi_\theta(y_{i,k} \mid x_{i,k})}{\pi_{\theta_{\text{old}}}(y_{i,k} \mid x_{i,k})} \hat{A}_{i,k}, \text{clip}\left( \frac{\pi_\theta(y_{i,k} \mid x_{i,k})}{\pi_{\theta_{\text{old}}}(y_{i,k} \mid x_{i,k})}, \right. \right. \qquad (6)$$

$$\left. \left. 1-\epsilon, 1+\epsilon \right) \hat{A}_{i,k} \right) - \lambda_{\text{KL}} D_{\text{KL}}(\pi_\theta \| \pi_{\text{ref}}) \Bigg\} \right],$$

where $x_{i,k}$ denotes the context $(x_i, y_{i,<k})$ for the $k$-th token.

### 4.3. Theory: Info-Gain as a Necessary Resource

We summarize two key results proving that our turn-level info-gain reward is a rigorous measure of information progress. Full proofs are provided in Appendix B.

**Theorem 1** (Equivalence to Mutual Information). *Let $H_t$ be the interaction history, $O_t$ the feedback, and $A_{t+1}$ the subsequent action. Defining the marginal policy $\pi_\theta(\cdot \mid H_t) \triangleq \mathbb{E}_{O_t \sim P(\cdot|H_t)}\left[\pi_\theta(\cdot \mid H_t, O_t)\right]$, the turn-level info-gain reward $r_t^{\text{info}}$ defined in Eq. 2 satisfies:*

$$\mathbb{E}[r_t^{\text{info}}] = I_\theta(O_t; A_{t+1} \mid H_t).$$

Theorem 1 equates the info-gain reward to the conditional mutual information between feedback and actions, formalizing the directed information flow that drives decisions.

**Theorem 2** (Necessity for Task Success). *Consider a task with a hidden intent $Z \sim \text{Unif}([M])$ and a terminal reward $R^{\text{ext}} = \mathbb{I}[\hat{Z} = Z]$. Achieving a success probability $\mathbb{P}(R^{\text{ext}} = 1) \geq 1 - \delta$ requires the cumulative info-gain reward to satisfy the lower bound:*

$$\mathbb{E}\left[ \sum_{t=0}^{T-1} r_t^{\text{info}} \right] \geq \log M - h(\delta) - \delta \log(M-1),$$

*where $h(\delta)$ is the binary entropy (in nats).*

Theorem 2 shows that high task success requires accumulating a minimum amount of information progress.

## 5. Experiments

Our experiments aim to answer the following research questions: **RQ1 (Performance):** How effectively does InfoPO facilitate task completion and learning efficiency across benchmarks? **RQ2 (Mechanism):** How do turn-level info-gain rewards and variance-gated fusion contribute to interaction quality and learning stability? **RQ3 (Generalization):** How robustly does InfoPO generalize to unseen interaction purposes and varying user simulator conditions?

### 5.1. Experimental Setup

**Benchmarks and Metrics.** We evaluate InfoPO on three representative multi-turn, user-centric benchmarks that challenge an agent's dual-interaction capabilities: eliciting latent intent via conversation while executing environment-grounded actions. (1) **UserGym** (Qian et al., 2025c): A suite of eight unified gym environments covering diverse interaction paradigms such as travel planning, preference persuasion, and goal inference. It features a standardized [Action-Search-Answer] interface, forcing the agent to coordinate tool usage with information-seeking dialogue. Following Qian et al. (2025c), we report the success rate or accumulated reward across eight tasks, including three held-out settings to test cross-purpose generalization. (2) **ColBench** (Zhou et al., 2025): A collaborative programming benchmark where the agent refines technical requirements and generates Python code through iterative discussion. Performance is measured by the fraction of hidden unit tests passed (Pass) and the task completion rate (Succ.). (3) $\tau^2$**-Bench** (Barres et al., 2025): A complex dual-control task involving airline, retail, and telecom domains. It requires high-level coordination as both the agent and user can modify the shared world state. We report the average success rate over 4 independent runs (Avg@4). Detailed task descriptions are provided in Appendix A.

**Baselines.** We compare InfoPO against the following baselines: (i) **UserRL** (Qian et al., 2025c), a representative user-centric multi-turn training framework. We adopt its strongest *Equalized/R2G* setting, which applies length-normalized rewards (*Equalized*) and future-reward-based advantage estimation (*R2G*) to the entire interaction sequence. (ii) **RAGEN** (Wang et al., 2025), which focuses on training stability through variance-based trajectory filtering and decoupled clipping. (iii) **Search-R1** (Jin et al., 2025), which optimizes search-based reasoning via retrieved-token masking. (iv) **ReAct** (Yao et al., 2022) and **Reflexion** (Shinn et al., 2023), serving as non-training prompting baselines to quantify the necessity of policy optimization.

*Table 1.* Experimental results on UserGym, ColBench, and $\tau^2$-Bench. The grey rows represent our proposed method and its ablation versions. **Best** and second-best results are marked within each model group (Closed-Source/Qwen2.5/Qwen3). Environment abbreviations: Func.: FunctionGym, Pers.: PersuadeGym, Inten.: IntentionGym, Tele.: TelepathyGym, Telec.: Telecom, Air.: Airline. Succ. denotes Success rate.

| Type | Method | UserGym | | | | | | | | ColBench | | $\tau^2$-Bench | | | Avg |
|---|---|---|---|---|---|---|---|---|---|---|---|---|---|---|---|
| | | Travel | Func. | Pers. | Tau | Turtle | *Search* | *Inten.* | *Tele.* | Pass | Succ. | Telec. | Retail | Air. | |
| *Closed-Source Model* | | | | | | | | | | | | | | | |
| Prompting | Gemini-3-Flash | 0.574 | **0.423** | **0.695** | **0.167** | 0.153 | **0.968** | 1.718 | **0.829** | 0.515 | 0.382 | **0.469** | **0.619** | **0.558** | **0.621** |
| Prompting | GPT-4.1 | 0.554 | 0.051 | 0.599 | 0.109 | **0.267** | 0.480 | **1.867** | 0.732 | **0.529** | **0.403** | 0.388 | 0.544 | 0.400 | 0.533 |
| Prompting | GPT-4o-mini | **0.596** | 0.089 | 0.683 | 0.091 | 0.117 | 0.568 | 1.277 | 0.512 | 0.463 | 0.342 | 0.113 | 0.400 | 0.175 | 0.417 |
| *Qwen2.5-7B-Instruct* | | | | | | | | | | | | | | | |
| Prompting | Qwen2.5 | 0.441 | 0.026 | 0.289 | 0.000 | 0.062 | 0.376 | 1.254 | 0.292 | 0.242 | 0.140 | 0.144 | 0.131 | 0.075 | 0.267 |
| Prompting | ReAct | 0.452 | 0.064 | 0.325 | 0.037 | 0.078 | 0.354 | 1.378 | 0.316 | 0.238 | 0.135 | 0.131 | 0.138 | 0.100 | 0.288 |
| Prompting | Reflexion | 0.445 | 0.052 | 0.312 | 0.022 | 0.074 | 0.364 | 1.320 | 0.305 | 0.276 | 0.168 | 0.138 | 0.131 | 0.075 | 0.283 |
| RL Training | RAGEN | 0.538 | 0.124 | **0.548** | 0.000 | 0.148 | 0.446 | 1.815 | 0.416 | 0.449 | 0.348 | 0.175 | 0.175 | 0.150 | 0.410 |
| RL Training | Search-R1 | 0.565 | 0.113 | 0.412 | 0.043 | 0.154 | 0.435 | 1.805 | 0.416 | 0.457 | 0.352 | 0.156 | 0.150 | 0.100 | 0.397 |
| RL Training | UserRL | 0.546 | 0.115 | 0.444 | 0.048 | 0.152 | 0.429 | 1.826 | 0.424 | 0.436 | 0.327 | 0.138 | 0.169 | 0.075 | 0.395 |
| RL Training | InfoPO w/o std | 0.565 | 0.142 | 0.498 | 0.035 | 0.065 | 0.455 | 1.845 | 0.452 | 0.395 | 0.298 | 0.169 | 0.162 | 0.125 | 0.400 |
| RL Training | InfoPO w/o Gate | 0.542 | 0.125 | 0.465 | 0.082 | 0.042 | 0.432 | 1.810 | 0.465 | 0.466 | 0.368 | 0.150 | 0.156 | 0.100 | 0.400 |
| RL Training | InfoPO w/o $R_{ext}$ | 0.485 | 0.055 | 0.352 | 0.035 | 0.015 | 0.385 | 1.450 | 0.325 | 0.285 | 0.342 | 0.112 | 0.125 | 0.088 | 0.312 |
| RL Training | **InfoPO (Ours)** | **0.588** | **0.167** | 0.535 | **0.091** | **0.178** | **0.480** | **1.892** | **0.488** | **0.534** | **0.426** | **0.181** | **0.188** | **0.163** | **0.455** |
| *Qwen3-4B* | | | | | | | | | | | | | | | |
| Prompting | Qwen3 | 0.277 | 0.026 | 0.452 | 0.006 | 0.071 | 0.444 | 1.660 | 0.488 | 0.272 | 0.153 | 0.106 | 0.225 | 0.062 | 0.326 |
| Prompting | ReAct | 0.279 | 0.053 | 0.476 | 0.005 | 0.147 | 0.435 | 1.782 | 0.454 | 0.269 | 0.145 | 0.094 | 0.225 | 0.088 | 0.342 |
| Prompting | Reflexion | 0.291 | 0.045 | 0.474 | 0.035 | 0.124 | 0.438 | 1.717 | 0.501 | 0.303 | 0.184 | 0.100 | 0.200 | 0.075 | 0.345 |
| RL Training | RAGEN | 0.477 | 0.104 | 0.511 | 0.006 | 0.117 | 0.714 | 1.572 | 0.488 | 0.479 | 0.361 | 0.137 | 0.231 | 0.088 | 0.407 |
| RL Training | Search-R1 | 0.482 | 0.103 | 0.425 | 0.059 | 0.113 | 0.702 | 1.711 | 0.412 | 0.467 | 0.355 | 0.118 | 0.234 | 0.088 | 0.405 |
| RL Training | UserRL | 0.538 | 0.095 | 0.507 | 0.053 | 0.121 | 0.697 | 1.732 | 0.520 | 0.468 | 0.342 | 0.100 | 0.163 | 0.075 | 0.416 |
| RL Training | InfoPO w/o std | 0.512 | 0.102 | 0.518 | 0.075 | 0.118 | 0.795 | 1.812 | 0.485 | 0.498 | 0.395 | 0.142 | 0.215 | 0.088 | 0.443 |
| RL Training | InfoPO w/o Gate | 0.548 | 0.085 | 0.482 | 0.062 | 0.115 | 0.758 | 1.765 | 0.460 | 0.475 | 0.372 | 0.131 | 0.198 | 0.075 | 0.425 |
| RL Training | InfoPO w/o $R_{ext}$ | 0.352 | 0.045 | 0.385 | 0.032 | 0.088 | 0.425 | 1.512 | 0.382 | 0.345 | 0.285 | 0.095 | 0.152 | 0.055 | 0.319 |
| RL Training | **InfoPO (Ours)** | **0.589** | **0.115** | **0.556** | **0.097** | **0.154** | **0.849** | **1.862** | **0.542** | **0.553** | **0.439** | **0.156** | **0.244** | **0.100** | **0.481** |

**Training details.** We use Qwen2.5-7B-Instruct (Qwen et al., 2025) and Qwen-3-4B (Yang et al., 2025a) as our base models. To evaluate the agent's intrinsic capacity to discover interaction strategies purely from reinforcement signals, all RL training is conducted directly without any SFT cold-start. We use group-based rollouts with $n=5$ samples per prompt. For multi-turn interaction, we cap the maximum number of turns to $16/10/50$ for UserRL, Col-Bench, and $\tau^2$-Bench, respectively. We set the batch size to $64/64/32$ for UserGym, ColBench, and $\tau^2$-Bench, and use a maximum rollout sequence length of 32,768 tokens. Across all benchmarks, we use GPT-4o-mini (Hurst et al., 2024) (temperature 0.7) as the default user simulator to balance high-fidelity interaction with computational efficiency and API throughput requirements. We follow the official train/test splits provided by each benchmark for both training and evaluation. All experiments are conducted on $4\times$ NVIDIA A800 GPUs (80GB). Other hyperparameter details are provided in the Appendix B.1. And we also discuss the compute overhead of the counterfactual evaluation in Appendix B.2; in our runs, the wall-clock cost is generally below $2\times$ (around $1.63\times$ on average).

## 5.2. RQ1: Overall Performance

Table 1 summarizes the final performance across three interactive benchmarks. On the Qwen2.5-7B-Instruct backbone, InfoPO achieves the strongest overall results among open-source RL baselines. In UserGym, InfoPO improves upon the strongest baseline in 7 out of 8 sub-environments, with particularly significant gains in cross-purpose generalization settings that require resolving underspecified goals (e.g., Search: 0.480 vs. 0.446; Intention: 1.892 vs. 1.826; Telepathy: 0.488 vs. 0.424). For code-centric tasks in ColBench, InfoPO yields clear improvements on both technical metrics (Pass: 0.534 vs. 0.457; Success: 0.426 vs. 0.352), slightly exceeding the performance of GPT-4.1 (0.529/0.403).

The superior efficiency of InfoPO is directly attributable to its improved credit assignment under non-informative feedback. As shown in Table 2, a significant portion of roll-out groups exhibits zero outcome variance during the initial training phase (31.3%–38.4% in UserGym/ColBench, and up to 76.3% in $\tau^2$-Bench). While standard group-relative advantages become brittle or provide near-zero gradients in these scenarios, InfoPO's info-gain reward serves as a dense training scaffold to bootstrap policy improvement. Consistent with this, the training dynamics in Figure 3 demonstrate that InfoPO initiates optimization earlier and reaches higher reward levels with reduced oscillations compared to base-

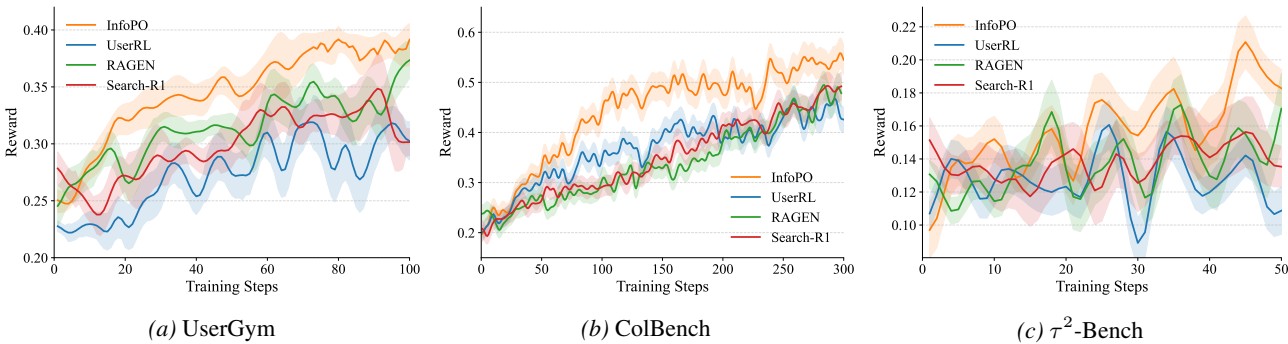

*Figure 3.* Extrinsic reward curves during training on (a) UserGym, (b) ColBench, and (c) $\tau^2$-Bench. Solid lines and shaded regions represent mean $\pm$ std across three seeds.

lines. Qualitative inspection of interaction traces (see Figure 12, 13 and 14 in Appendix D) reveals that InfoPO-trained agents exhibit structured and proactive strategies, such as resolving intent ambiguity through early-turn clarification, reflecting a form of behavioral maturation.

*Table 2.* Percentage of rollout groups with zero outcome variance during the initial training phase.

| Model | UserGym | ColBench | $\tau^2$-Bench |
|---|---|---|---|
| Qwen2.5-7B-Inst. | 0.384 | 0.313 | 0.763 |
| Qwen3-4B | 0.421 | 0.345 | 0.812 |

On the long-horizon $\tau^2$-Bench, InfoPO maintains its competitiveness by matching or improving upon the best open-source baselines across all task families (Telecom: 0.181; Retail: 0.188; Air: 0.150). Considering $\tau^2$-Bench's extreme interaction horizon (often $> 30$ turns) and severe data scarcity (only 178 tasks), the steady improvement from a base instruction model validates InfoPO's effectiveness.

### 5.3. RQ2: InfoPO Mechanism Analysis

**Ablations.** To isolate the contributions of InfoPO's core designs, we evaluate three variants: InfoPO **w/o** $R_{\text{ext}}$ (pure information gain), **w/o Gate** (fixed weighting without variance-based gating), and **w/o std** (removing info-gain normalization). Table 1 and Figure 4 summarize final performance ($J_f$) together with stability and interaction diagnostics (e.g., $\Delta_{\text{bf}}$, $P_{\text{cr}}$, and length-related statistics; see Appendix A.4 for formal definitions). Removing extrinsic supervision (**w/o** $R_{\text{ext}}$) leads to a consistent and substantial drop across nearly all tasks, underscoring that information seeking alone is insufficient without task-level grounding. Disabling dynamic gating (**w/o Gate**) primarily hurts training stability, manifested as larger best-to-final regression $\Delta_{\text{bf}}$ and higher collapse probability $P_{\text{cr}}$, validating that the gate is important for preventing late-stage objective drift when the policy transitions from early uncertainty reduction to outcome refinement. Finally, removing standardization (**w/o std**) degrades both performance and interaction behav-

ior: it not only lowers $J_f$ but also increases length sensitivity (higher $|\rho_{L,r}|$), indicating that group-relative normalization is critical for stabilizing turn-level credit assignment and preventing the information advantage from being dominated by a small number of high-magnitude but noisy turns.

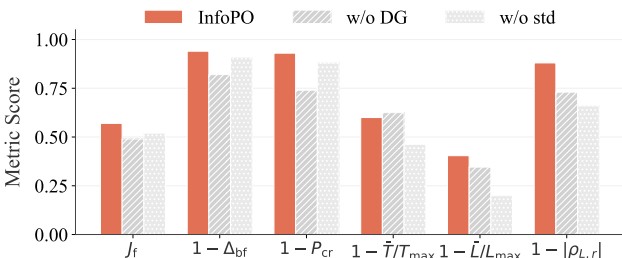

*Figure 4.* Mechanism diagnostics under ablations (aggregated across tasks). $J_f$ denotes final extrinsic performance; $\Delta_{\text{bf}}$ is the best-to-final drop (late-training instability); $P_{\text{cr}}$ is the probability of training collapse; $\bar{T}$ and $\bar{L}$ are the average interaction turns and response length; and $\rho_{L,r}$ is the correlation between response length and extrinsic reward (a proxy for length-based reward hacking). All metrics are converted to "higher-is-better" scores in the figure; formal definitions are in Appendix A.

**Interaction Dynamics.** A primary concern in multi-objective RL is whether information rewards merely incentivize longer, more repetitive interactions. To investigate this, we track interaction turns and response lengths throughout training (Figure 5a). In UserGym and ColBench, InfoPO exhibits an emergent "explore-then-consolidate" pattern: it temporarily increases the number of turns in early training to reduce intent uncertainty, while steadily shortening each turn's response length. In contrast, baselines like UserRL monotonically shrink both turns and length from the start, often collapsing to "short-horizon" behaviors that prematurely commit to actions without sufficient information. Figure 5b further reveals that while the absolute info-gain signal increases as the agent becomes more inquisitive, its relative contribution to the final advantage decreases via the variance gate. This confirms that InfoPO utilizes interaction as a strategic resource—expanding it only when necessary

to acquire task-critical information and consolidating into efficient execution as outcome-based learning takes over.

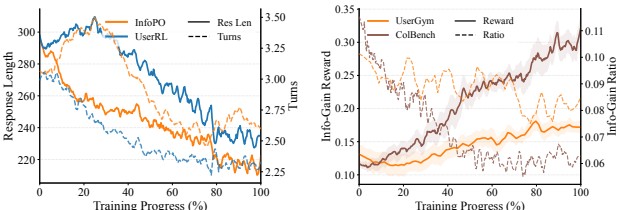

*(a)* Turns vs. Response Length   *(b)* Info-gain Reward Dynamics

*Figure 5.* Interaction dynamics and reward signals. (a) Turns vs. response length; (b) Absolute info-gain (solid) and its advantage contribution ratio (dashed). See Appendix B.4 for more results.

Interestingly, in the high-horizon $\tau^2$-Bench domain (up to 50 turns), InfoPO shifts toward a "pruning-first" strategy, with both interaction turns and response lengths decreasing from the onset of training (see Appendix B.4 for detailed curves). Unlike the shorter tasks in UserGym, the base model's initial trajectories in $\tau^2$-Bench are already excessively redundant due to the environment's complexity. In this regime, InfoPO's turn-level credit assignment immediately identifies and penalizes non-informative actions, directing the policy toward more concise and goal-oriented behaviors without the need for an initial exploration expansion. This context-aware behavior demonstrates that InfoPO adaptively modulates interaction depth based on the inherent difficulty and information density of the task environment.

**Per-turn credit assignment.** Figure 6 shows the distribution of the info-gain reward over turns during training on UserGym, averaged over valid turns at each step. Early on, rewards are spread across the dialogue; as training progresses, they concentrate on the first few turns, indicating an emergent "clarify-then-act" behavior. Specifically, successful policies learn to place the most discriminative questions at the beginning to elicit high-information feedback, after which intrinsic rewards naturally decay as the agent executes a specified task. This trend is not imposed by any heuristic: under our counterfactual objective, a turn is rewarded only when the observed feedback induces a measurable shift in the policy's subsequent decision distribution. Consequently, InfoPO learns where to ask rather than simply to ask more, yielding a self-organized progression that prioritizes early uncertainty reduction as a precursor to task success.

**Trajectory-quality diagnostics.** A natural concern is whether the counterfactual signal only measures arbitrary distribution shift, especially when sampled trajectories contain low-quality actions. We therefore analyze $\tau^2$-Bench, the most failure-prone benchmark due to its long horizon, tool use, and sparse rewards. We label each action as *good*, *meaningless*, or *informative failure*, and measure the observation sensitivity that defines $r_t^{\text{info}}$. Good next actions are

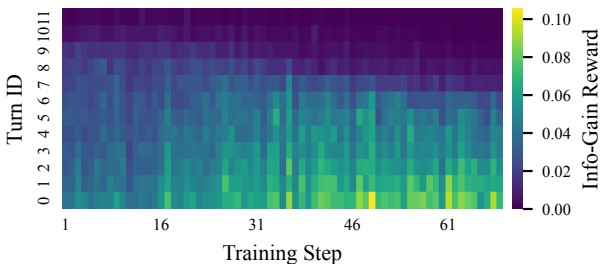

*Figure 6.* Heatmap of turn-level info-gain rewards during training.

substantially more sensitive to the observation than meaningless ones (0.275 vs. 0.098; tool calls only: 0.553 vs. 0.110). When grouping by the current action, good actions also induce much higher next-step sensitivity than meaningless actions (mean 0.380 vs. 0.017). These results suggest that useful transitions naturally receive stronger counterfactual credit, while uninformative actions receive little signal. Full definitions, statistical tests, and transition plots are provided in Appendix B.5.

**Reward-hacking analysis.** Another concern is that an intrinsic information reward may encourage generic open-ended questions that elicit long but task-irrelevant responses. We test this by comparing *targeted clarifications* (questions about predefined task-relevant fields or diagnostic slots) with *generic open questions*. Under InfoPO, the top-10% high-$r^{\text{info}}$ turns are much more concentrated on targeted clarifications than generic open questions (58.8% vs. 12.4%). The correlation between response length and $r^{\text{info}}$ is weak (0.157), and a length-matched comparison still favors targeted clarifications. In contrast, removing the variance gate makes generic open questions much more frequent and increases both interaction length and response length. This supports the role of variance-gated fusion in keeping information seeking aligned with task-relevant uncertainty reduction. Detailed results are reported in Appendix B.6.

### 5.4. RQ3: Generalization

**Environment generalization.** To evaluate whether InfoPO generalizes beyond user-centric environments, we conduct experiments on **Sokoban** and **WebShop** following the protocols in RAGEN. Sokoban is a multi-turn task requiring planning in a grid-world environment, while WebShop requires grounding in a realistic web interface. Using Qwen2.5-1.5B-Instruct as the base model, we observe that InfoPO successfully mitigates the "Echo Trap" failure mode – a common collapse in standard GRPO where policies regress to repetitive, locally-rewarded templates once rollout groups fail to reach the goal (Wang et al., 2025). As shown in Figure 7, InfoPO maintains a stable upward trend in success rate where baselines collapse. This demonstrates

that InfoPO's effectiveness goes beyond specific user-centric scenarios, framing general multi-turn interaction as a fundamental process of active uncertainty reduction.

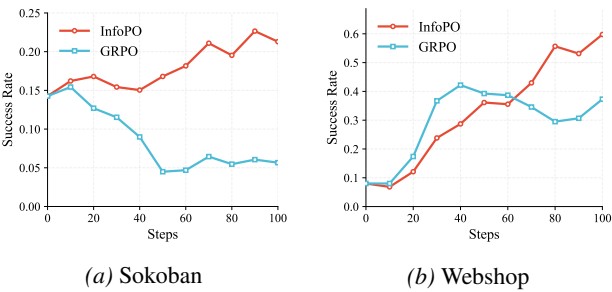

*(a) Sokoban*      *(b) Webshop*

*Figure 7.* Success rates on environment-interactive tasks. InfoPO maintains a stable upward learning trend on Sokoban and Web-Shop, demonstrating its robust generalization beyond user-centric scenarios.

**User generalization.** Since all training runs use GPT-4o-mini as the simulated user for cost efficiency, we further examine whether InfoPO's performance is sensitive to the user employed at test time. We evaluate InfoPO (Qwen-2.5-7B) under three user configurations: (**Base**) GPT-4o-mini with the official benchmark prompt; (**Optimized Prompt, OP**) the same model using constrained instructions to enforce protocol compliance and minimize unforced errors in consistency or tool-call execution. (See Appendix E for prompts); and (**Optimized Model, OM**) a stronger model, GPT-4.1, using the official prompt. We report the average scores across all metrics for each benchmark in Table 3.

*Table 3.* Comparison of different simulated users at test time. Base uses GPT-4o-mini with default prompts, OP uses optimized prompts, and OM uses a stronger model (GPT-4.1).

| User | UserGym | ColBench | $\tau^2$-Bench |
|---|---|---|---|
| Base | 0.552 | 0.480 | 0.173 |
| Optim. Prompt (OP) | 0.563 | 0.483 | 0.203 |
| Optim. Model (OM) | 0.558 | 0.502 | 0.215 |

Overall, stronger user simulators yield consistent but task-dependent effects. OM brings the largest gains on $\tau^2$-Bench, where coordination over tool calls is critical and user mistakes can cause failures unrelated to the agent policy. Col-Bench shows smaller yet generally positive improvements, suggesting moderate sensitivity to user reliability. In contrast, UserGym shows mixed effects, consistent with its design that enforces realistic user behaviors (e.g., progressive disclosure, resistance to persuasion, and strict non-hallucination), which stronger simulators follow more faithfully and can therefore make tasks harder.

## 6. Conclusion

We introduced **InfoPO**, an information-driven policy optimization method for user-centric multi-turn agents that treats interaction as active uncertainty reduction. InfoPO provides dense, turn-level credit assignment by measuring how each observation counterfactually changes the policy's next-action distribution, and keeps this intrinsic signal aligned with task success via a variance-gated fusion with outcome advantages. This yields a task-agnostic yet scalable learning signal for long-horizon interaction where group-relative updates often stall under sparse or non-discriminative rewards. Across three interactive benchmarks, InfoPO consistently improves performance, sample efficiency, and training stability over strong prompting and RL baselines, and further generalizes to user-simulator shifts and non-user environment interaction tasks.

## Impact Statement

InfoPO improves the reliability of user-facing AI agents by encouraging them to ask clarification questions before taking action. This may help reduce errors in multi-turn interactions, but more capable agents may also raise privacy and safety concerns if they infer sensitive information or misunderstand user intent. We believe such systems should be deployed with appropriate safeguards and user oversight. Our experiments are conducted only in simulated benchmark environments.

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

# A. InfoPO Pseudo-Code

Algorithm 1 outlines the complete training procedure of Information-Driven Policy Optimization (InfoPO). In each training iteration, the algorithm first samples a group of $G$ trajectories by interacting with the environment or user simulator up to a horizon $T$ (Line 2). For each trajectory, it evaluates the trajectory-level external reward $R^{ext}$ (Line 3). Crucially, the turn-level information-gain reward $r_t^{info}$ is computed via the InfoGainPerTurn function (Lines 4, 11-20). This function iterates through each valid turn, comparing the policy's log-probability of generating the realized next action given the factual context (with the real observation $o_t$) against a counterfactual context where the observation is replaced by a mask placeholder $\emptyset$.

After obtaining both reward signals, they are standardized within the rollout group to compute the external advantage $A^{ext}$ and the info-gain advantage $A^{info}$ (Lines 5-6). To adaptively balance task completion and information seeking, InfoPO calculates a variance-based gate $g$ (Line 7), which amplifies the influence of the info-gain advantage when the standard deviation of external rewards within the group is small (i.e., when feedback is sparse or non-discriminative). Finally, the unified token-level advantage $\hat{A}$ is computed, broadcast to the corresponding response tokens, and used to update the policy $\pi_\theta$ via a standard PPO clipped objective, regularized by a KL-divergence penalty against the reference model $\pi_{ref}$ (Lines 8-9).

---

**Algorithm 1** INFOPO

---

**Require:** policy $\pi_\theta$, reference $\pi_{\text{ref}}$, env $\mathcal{E}$; group size $G$, horizon $T$; mask placeholder $\varnothing$, weight $\beta$, gate temperature $\tau$, PPO clip $\epsilon$
 1: **for** each iteration **do**
 2:     Sample $G$ trajectories $\{\tau_i\}_{i=1}^G$ of $T$ turns from $\pi_\theta$ in $\mathcal{E}$, record turn boundaries.
 3:     $R_i^{\text{ext}} \leftarrow \text{ScoreExt}(\tau_i)$
 4:     $\{r_{i,t}^{\text{info}}\}_{t=1}^T \leftarrow \text{InfoGainPerTurn}(\tau_i, \pi_\theta, \pi_{\text{ref}}, \varnothing)$
 5:     $A^{\text{ext}} \leftarrow \text{GroupNormExt}(\{R_i^{\text{ext}}\}_{i=1}^G)$
 6:     $A^{\text{info}} \leftarrow \text{GroupNormInfo}(\{r_{i,t}^{\text{info}}\}_{i,t})$
 7:     $g \leftarrow \sigma(-\text{Std}_g(\{R_i^{\text{ext}}\})/\tau)$
 8:     $\hat{A} \leftarrow A^{\text{ext}} + \beta \cdot g \cdot A^{\text{info}}$, broadcast to response tokens
 9:     Update $\theta$ using $\hat{A}$ and KL-to-$\pi_{\text{ref}}$ regularization.
10: **end for**
11: **function** InfoGainPerTurn($\tau, \pi_\theta, \pi_{\text{ref}}, \varnothing$)
12:     Initialize $r_t^{\text{info}} \leftarrow 0$ for $t = 1..T$
13:     **for** $t = 1$ to $T - 1$ **do**
14:         Let $c_t$ be context up to (action$_t$, obs$_t$).
15:         $\ell^{\text{post}} \leftarrow \log \pi_\theta(a_{t+1} \mid h_t, o_t)$,
16:         $\ell^{\text{prior}} \leftarrow \log \tilde{\pi}_\theta(a_{t+1} \mid h_t)$,
17:         $r_t^{\text{info}} \leftarrow \ell^{\text{post}} - \ell^{\text{prior}}$. {attributed to turn $t$}
18:     **end for**
19:     **return** $\{r_t^{\text{info}}\}_{t=1}^T$
20: **end function**

---

# B. Proofs of Theoretical Results

### B.1. Proof of Theorem 1

We consider a turn-based interaction of horizon $T$. Let $H_0$ denote the initial context (prompt, system message, etc.). For each turn $t \in \{0, \ldots, T - 1\}$, the environment (or user simulator) produces feedback $O_t$, and the agent then produces the next action segment $A_{t+1}$. Define the (pre-feedback) history random variable

$$H_t \triangleq (H_0, A_{1:t}, O_{0:t-1}),$$

where $A_{1:t} = (A_1, \ldots, A_t)$ and $O_{0:t-1} = (O_0, \ldots, O_{t-1})$. The policy is *causal*:

$$P_\theta(A_{t+1} \mid H_t, O_t) = \pi_\theta(A_{t+1} \mid H_t, O_t).$$

We further define the *marginal (prior) policy* by averaging over the next feedback:

$$\pi_\theta(a \mid h_t) \triangleq \mathbb{E}_{O_t \sim P(\cdot \mid h_t)}\big[\pi_\theta(a \mid h_t, O_t)\big],$$

where the expectation is w.r.t. the environment's conditional distribution of $O_t$ given $H_t = h_t$. (For continuous variables, replace sums by integrals; the derivation remains identical.) InfoPO intrinsic reward is defined as

$$r_{\text{info}}(t) \triangleq \log \pi_\theta(A_{t+1} \mid H_t, O_t) - \log \pi_\theta(A_{t+1} \mid H_t).$$

**Part I: Per-turn expectation equals conditional mutual information.** Taking expectation over the joint distribution of $(H_t, O_t, A_{t+1})$ induced by the environment and policy,

$$\begin{aligned}
\mathbb{E}[r_{\text{info}}(t)] &= \mathbb{E}_{H_t, O_t, A_{t+1}}\left[\log \frac{\pi_\theta(A_{t+1} \mid H_t, O_t)}{\pi_\theta(A_{t+1} \mid H_t)}\right] \\
&= \sum_{h_t} P(h_t) \sum_{o_t} P(o_t \mid h_t) \sum_a \pi_\theta(a \mid h_t, o_t) \log \frac{\pi_\theta(a \mid h_t, o_t)}{\pi_\theta(a \mid h_t)}.
\end{aligned} \tag{7}$$

Note that the conditional joint distribution given $H_t = h_t$ factorizes as

$$P(o_t, a \mid h_t) = P(o_t \mid h_t)\pi_\theta(a \mid h_t, o_t).$$

Substituting into (7), we obtain

$$\mathbb{E}[r_{\text{info}}(t)] = \sum_{h_t} P(h_t) \sum_{o_t, a} P(o_t, a \mid h_t) \log \frac{P(o_t, a \mid h_t)}{P(o_t \mid h_t)\, P(a \mid h_t)}.$$

By the definition of conditional mutual information,

$$I_\theta(O_t; A_{t+1} \mid H_t) \triangleq \sum_{h_t} P(h_t) \sum_{o_t, a} P(o_t, a \mid h_t) \log \frac{P(o_t, a \mid h_t)}{P(o_t \mid h_t)\, P(a \mid h_t)},$$

we conclude

$$\mathbb{E}[r_{\text{info}}(t)] = I_\theta(O_t; A_{t+1} \mid H_t).$$

**Part II: Cumulative InfoPO intrinsic equals directed information.** Define $O^t \triangleq O_{0:t}$ and $A^t \triangleq A_{1:t}$. Following Massey (1990), the *directed information* from feedback to actions is defined as the sum of conditional mutual informations:

$$I_\theta(O^{T-1} \to A^T \mid H_0) \triangleq \sum_{t=0}^{T-1} I_\theta(O_t; A_{t+1} \mid H_0, A^t, O^{t-1}).$$

Recalling that the history is $H_t = (H_0, A^t, O^{t-1})$, conditioning on $(H_0, A^t, O^{t-1})$ is exactly equivalent to conditioning on $H_t$. Therefore, each summand simplifies directly:

$$I_\theta(O_t; A_{t+1} \mid H_0, A^t, O^{t-1}) = I_\theta(O_t; A_{t+1} \mid H_t).$$

Summing over $t$ and applying the result from Part I:

$$I_\theta(O^{T-1} \to A^T \mid H_0) = \sum_{t=0}^{T-1} \mathbb{E}[r_{\text{info}}(t)] = \mathbb{E}\left[\sum_{t=0}^{T-1} r_{\text{info}}(t)\right].$$

This completes the proof. ∎

## B.2. Proof of Theorem 2

**Problem model.** Let $Z \sim \mathrm{Unif}(\{1, \ldots, M\})$ be a hidden user intent (or goal) that the agent must infer through interaction. The agent interacts for $T$ turns and outputs actions $A^T = (A_1, \ldots, A_T)$. A terminal estimator outputs $\hat{Z} = \varphi(A^T)$ and the external reward is

$$R^{\mathrm{ext}} = \mathbb{I}[\hat{Z} = Z].$$

Assume $Z$ is independent of the initial context $H_0$, hence $H(Z \mid H_0) = \log M$ (natural logs; units are nats). Suppose the success probability satisfies

$$\mathbb{P}(\hat{Z} = Z) \geq 1 - \delta.$$

**Step 1: High success implies large mutual information about $Z$ (Fano).** By Fano's inequality, for any estimator $\hat{Z}$ of $Z$,

$$H(Z \mid \hat{Z}) \leq h(\delta) + \delta \log(M - 1),$$

where $h(\delta) \triangleq -\delta \log \delta - (1 - \delta) \log(1 - \delta)$ is the binary entropy (nats). Since $\hat{Z} = \varphi(A^T)$ is a deterministic function of $A^T$, conditioning on $A^T$ is at least as informative as conditioning on $\hat{Z}$, i.e.,

$$H(Z \mid A^T, H_0) \leq H(Z \mid \hat{Z}, H_0) = H(Z \mid \hat{Z}).$$

Therefore,

$$\begin{aligned} I(Z; A^T \mid H_0) &= H(Z \mid H_0) - H(Z \mid A^T, H_0) \\ &\geq \log M - \big(h(\delta) + \delta \log(M - 1)\big). \end{aligned} \tag{8}$$

**Step 2: Bounding latent intent information via feedback-to-action flow.** To establish the necessity of information gain, we derive an upper bound on $I(Z; A^T \mid H_0)$ by decomposing the information acquired at each interaction turn. By the chain rule of mutual information:

$$I(Z; A^T \mid H_0) = \sum_{t=0}^{T-1} I(Z; A_{t+1} \mid A^t, H_0). \tag{9}$$

For each turn $t$, we bound the information $A_{t+1}$ contains about $Z$ by introducing the feedback history $O^t = (O^{t-1}, O_t)$. Using the monotonicity and the chain rule of mutual information:

$$\begin{aligned} I(Z; A_{t+1} \mid A^t, H_0) &\leq I(Z, O^t; A_{t+1} \mid A^t, H_0) \\ &= I(O^t; A_{t+1} \mid A^t, H_0) + I(Z; A_{t+1} \mid A^t, O^t, H_0). \end{aligned} \tag{10}$$

The second term $I(Z; A_{t+1} \mid A^t, O^t, H_0)$ vanishes due to the *Causal Markov Property* of the agent's policy: given the prompt and the observation history $(H_0, A^t, O^t)$, the action $A_{t+1}$ is generated solely by $\pi_\theta$ and does not depend on the latent intent $Z$ directly. Substituting this back into Eq. 9, we obtain:

$$I(Z; A^T \mid H_0) \leq \sum_{t=0}^{T-1} I(O^t; A_{t+1} \mid A^t, H_0). \tag{11}$$

To isolate the contribution of the *new* feedback $O_t$, we expand $I(O^t; A_{t+1} \mid A^t, H_0)$ as:

$$I(O^t; A_{t+1} \mid A^t, H_0) = \underbrace{I(O_t; A_{t+1} \mid A^t, O^{t-1}, H_0)}_{\text{Innovation (InfoPO)}} + \underbrace{I(O^{t-1}; A_{t+1} \mid A^t, H_0)}_{\text{History Dependency}}. \tag{12}$$

In a well-designed interaction, information about $Z$ should be extracted through the *innovation* provided by $O_t$ relative to the current history. Formally, for a causal feedback channel where $Z \to O^T \to A^T$ forms a sequence, the *Directed Data Processing Inequality* (Massey et al., 1990; Kim, 2008) states that the information about the source $Z$ is strictly bounded by the Directed Information flow:

$$I(Z; A^T \mid H_0) \leq I_\theta(O^{T-1} \to A^T \mid H_0) \triangleq \sum_{t=0}^{T-1} I(O_t; A_{t+1} \mid H_t), \tag{13}$$

where $H_t = (H_0, A^t, O^{t-1})$. This eliminates the redundant History Dependency term, as previous observations $O^{t-1}$ are already part of the conditioning context $H_t$ in an autoregressive agent.

**Step 3: Connecting to InfoPO intrinsic.** From Theorem 1, we know that the directed information matches the cumulative InfoPO reward:

$$I_\theta(O^{T-1} \to A^T \mid H_0) = \mathbb{E}\left[\sum_{t=0}^{T-1} r_{\text{info}}(t)\right].$$

Combining this equality with the directed data processing bound in Eq. 13 and the Fano lower bound in Eq. 8, we obtain:

$$\mathbb{E}\left[\sum_{t=0}^{T-1} r_{\text{info}}(t)\right] \geq I(Z; A^T \mid H_0) \geq \log M - h(\delta) - \delta \log(M - 1).$$

This completes the proof. ∎

## A. Task and Metrics Details

**Benchmarks.** We evaluate on three interactive benchmarks: UserGym (from UserRL), ColBench (from SWEET-RL), and $\tau^2$-Bench. They all require multi-turn interaction under a fixed environment interface, where the agent must balance information gathering (e.g., clarification or queries) with execution to complete the task.

### A.1. UserGym

**Interface and episode protocol.** UserGym is a unified suite of eight gym environments that share an `[Action--Search--Answer]` interface. Each episode starts from an underspecified request (or a latent rule/goal) and proceeds for at most $T_{\max} = 16$ turns in our setup. At each turn, the agent may execute an `Action` (environment-specific), optionally use `Search` when supported, and may terminate by emitting `Answer`.

**Eight gyms and what they test.** UserGym contains TravelGym, TurtleGym, FunctionGym, TauGym, PersuadeGym, IntentionGym, TelepathyGym, and SearchGym. TravelGym focuses on personalized travel planning under missing constraints; TurtleGym is a multi-turn reasoning game with incremental feedback; PersuadeGym tests argumentation toward a target stance; IntentionGym emphasizes disambiguating underspecified intent via targeted questions; TelepathyGym requires iterative hypothesis testing to identify a hidden entity; SearchGym tests search-and-answer style information seeking; TauGym is task-oriented tool use with user-provided details. FunctionGym is qualitatively different from the others: it is a latent mapping-rule inference environment rather than a user-in-the-loop dialogue task. The agent queries input–output mappings through `Action` and then answers held-out test cases, so interaction is driven by I/O probing rather than conversational user feedback.

**Train/test split for generalization.** We follow the standard protocol of training on five gyms and evaluating both in-domain and generalization to held-out interaction purposes, with IntentionGym, TelepathyGym, and SearchGym as held-out environments.

**UserGym metrics.** Depending on the gym, the evaluation score is either the success rate of the final decision (notably TravelGym and TauGym) or the accumulated reward over the episode (the remaining gyms). We report per-gym scores and the macro-average across the eight gyms.

### A.2. ColBench

**Tasks and interaction budget.** ColBench evaluates collaborative programming with a human simulator. In our experiments we use the Backend Programming setting only, where the agent iteratively refines an implementation of a Python function under incomplete specifications through multi-turn discussion and edits. We do not include the Frontend Design setting because it requires vision-language modeling to compare rendered pages, which is beyond the scope of this paper. Interactions are limited to at most 10 back-and-forth rounds in our setup.

**Metrics.** Backend Programming is evaluated by hidden unit tests (10 tests per task). We report Pass, defined as the average fraction of unit tests passed, and Succ., defined as the fraction of tasks that pass all unit tests.

## A.3. $\tau^2$-Bench

**Dual-control environment.** $\tau^2$-Bench evaluates customer-support style agents in simulated domains including airline, retail, and telecom. A defining feature is dual control: depending on the domain, the user simulator may also have tools and can modify the shared world state, so success depends on coordination between agent actions and user-side tool usage. We cap each episode at $T_{\max} = 50$ turns.

**Metric.** We report Avg@4, defined as the average task success rate over 4 independent runs per task instance, matching our experimental protocol.

## A.4. Additional diagnostics used in ablations

**Notation.** Let $b$ denote a benchmark (or gym), $s$ a random seed, and $u$ an evaluation checkpoint during training. Let $J_b(s, u)$ be the main evaluation score on $b$ for seed $s$ at checkpoint $u$. For an evaluation trajectory $\tau$, let $T(\tau)$ be the number of turns, and let $L_t(\tau)$ be the number of generated tokens in the agent response at turn $t$. We use $L_{\max} = 1024$ as the per-turn generation cap and the benchmark-specific $T_{\max}$ as the maximum turn budget.

**Final performance.** We define final performance on benchmark $b$ as the mean score at the final checkpoint:

$$J_f(b) \;=\; \frac{1}{|S|} \sum_{s \in S} J_b\big(s, u_{\text{final}}\big).$$

**Best-to-final drop.** To quantify late-training regression, we compute the best-to-final drop per seed and then average across seeds:

$$\Delta_{bf}(b) \;=\; \frac{1}{|S|} \sum_{s \in S} \left( \max_u J_b(s, u) \;-\; J_b\big(s, u_{\text{final}}\big) \right).$$

Larger values indicate stronger regression after reaching a good checkpoint.

**Collapse probability.** We mark a seed as collapsed if its final score falls below a fixed fraction of its own best checkpoint. With tolerance $\alpha = 0.5$,

$$\text{Collapse}(s) \;=\; \mathbb{I}\Big[ J_b\big(s, u_{\text{final}}\big) < \alpha \cdot \max_u J_b(s, u) \Big], \qquad P_{\text{cr}}(b) \;=\; \frac{1}{|S|} \sum_{s \in S} \text{Collapse}(s).$$

**Interaction length statistics.** We measure average turns per episode

$$\bar{T}(b) \;=\; \mathbb{E}_{\tau \sim \text{Eval}(b)}[T(\tau)], \qquad \text{and the utilization } \bar{T}(b)/T_{\max}.$$

We measure per-episode average response length

$$\bar{L}(b) \;=\; \mathbb{E}_{\tau \sim \text{Eval}(b)} \left[ \frac{1}{T(\tau)} \sum_{t=1}^{T(\tau)} L_t(\tau) \right], \qquad \text{and the utilization } \bar{L}(b)/L_{\max}.$$

**Length–reward correlation.** To probe whether higher rewards correlate with verbosity, we compute the Pearson correlation between episode-level average response length and the episode extrinsic reward:

$$\rho_{L,r}(b) \;=\; \text{Corr}\big(\ell(\tau), r(\tau)\big), \quad \ell(\tau) = \frac{1}{T(\tau)} \sum_{t=1}^{T(\tau)} L_t(\tau),$$

where $r(\tau)$ is the benchmark-defined extrinsic return for trajectory $\tau$.

*Table 4.* Key hyperparameters for training on UserGym, ColBench, and $\tau^2$-Bench environments.

| Parameter | UserGym | ColBench | $\tau^2$-Bench |
|---|---|---|---|
| *Info Gain Configuration* | | | |
| Intrinsic weight ($\beta_0$) | 0.5 | 0.3 | 0.1 |
| KL batch size | 4 | 4 | 2 |
| Gate temperature | 0.5 | 0.5 | 0.05 |
| *Data Configuration* | | | |
| Train batch size | 64 | 64 | 32 |
| Max prompt length | 1152 | 2048 | 8192 |
| Max response length | 8192 | 8192 | 16384 |
| *Actor Configuration* | | | |
| Learning rate | $3 \times 10^{-7}$ | $3 \times 10^{-7}$ | $1 \times 10^{-6}$ |
| PPO minibatch size | 16 | 16 | 16 |
| PPO microbatch size per gpu | 4 | 4 | 2 |
| KL loss coef | 0.001 | 0.001 | 0.001 |
| Entropy coefficient | 0.001 | 0.001 | 0.001 |
| *Rollout Configuration* | | | |
| Rollout engine | SGLang | SGLang | SGLang |
| Log prob micro batch size per gpu | 4 | 4 | 2 |
| Parallel rollouts ($n$) | 5 | 5 | 5 |
| Max turns | 16 | 10 | 50 |
| Response length (per turn) | 1024 | 1024 | 1024 |
| GPU memory utilization | 0.5 | 0.5 | 0.5 |
| *Training Configuration* | | | |
| GPUs per node | 4 | 4 | 4 |
| Total epochs | 2 | 2 | 10 |

# B. Experiment Details

## B.1. Training Details

The key hyperparameters used for training across all environments are summarized in Table 4. The parameter of $\beta$ controls the peak contribution of the information-gain advantage to the unified gradient. As shown in our sensitivity analysis in Appendix B.3, a range of 0.1 to 0.5 provides a stable training signal across diverse interaction paradigms. The choice of $T$ in the variance-gating function $g(\sigma_g^{ext}) = \sigma(-\sigma_g^{ext}/T)$ is based on the typical magnitude of the external reward variance ($\sigma_g^{ext}$) observed during the initial training phase.

The agent operates in a multi-turn conversational setting where inputs are formatted using the chat template of the underlying language model (e.g., Qwen's chat template). Each conversation begins with a system message defining the task, environment constraints, and available tools, followed by alternating user and assistant messages. User messages contain environment observations and feedback, while assistant messages contain the agent's actions or tool calls. The agent's output is generated autoregressively and can take two forms: (1) text responses for direct communication, and (2) tool calls in OpenAI function calling format, structured as JSON objects containing the tool name and parameters that are parsed and executed by the environment.

The action space is defined through a unified `interact_with_env` tool interface that abstracts environment-specific actions. All environments use a single tool that accepts a `choice` parameter indicating the action type and a `content` parameter containing the action details. The specific action space varies by environment: **UserGym** allows `["action", "answer", "search"]` for conversational interactions, information retrieval, and final responses; **ColBench** uses a single action type `["action"]` where the agent can ask clarification questions or provide Python code solutions prefixed with `"I WANT TO ANSWER:"`; $\tau^2$-**Bench** supports `["message", "tool_call", "done"]` for sending messages, executing domain-specific tools (e.g., database queries, booking operations), or terminating conversations, where tool calls can be specified in either JSON format or functional notation.

All tools follow the OpenAI function calling schema format, consisting of a `type` field set to `"function"`, a `function` object with `name`, `description`, and `parameters` fields. The `parameters` field uses JSON Schema to de-

fine the structure, including parameter types, constraints (e.g., `enum` for discrete choices), and descriptions. The `interact_with_env` tool schema includes a tool identifier, a natural language description of the tool's purpose, and a parameters object with `choice` (enumeration of valid action types) and `content` (string describing action details) as required fields. Tool schemas are provided to the model as part of the system prompt and are dynamically included in the chat template, enabling the model to generate properly formatted tool calls.

During training, we apply a loss mask to ensure that the model only learns from assistant-generated tokens, excluding system prompts, user messages, and special formatting tokens. For models using turn-based special tokens (e.g., Qwen's `<|im_start|>` and `<|im_end|>` tokens), we identify assistant turns by computing a cumulative sum of turn start tokens to create turn indicators, where odd-numbered turns (after the system message) correspond to assistant responses. The loss mask is set to 1 for all tokens in assistant turns and 0 elsewhere. For multi-turn trajectories, we use turn-level scoring where rewards are assigned to the last token of each assistant turn. For Qwen models, we apply a one-token shift to account for the newline character between the special token and the reward token, ensuring rewards are correctly associated with the final token of each response. The response mask, used for computing advantages and value estimates, follows the same pattern as the loss mask but is applied during the PPO update phase, ensuring that value function learning and policy updates are focused on the agent's actual responses rather than the input context.

## B.2. Compute Cost of InfoPO

InfoPO's per-turn info-gain reward $r_{i,t}^{\mathrm{info}}$ (Eq. 2) is computed by a counterfactual masking comparison under teacher forcing: for each valid turn, we evaluate the log-likelihood of the realized next action segment under the factual transcript (with $o_{i,t}$) and under the counterfactual transcript (with the placeholder $\varnothing$), and take their difference. This introduces additional policy forward evaluations but does not require any extra environment interactions. Concretely, let $N_g^{\mathrm{info}}$ denote the number of *valid* $(i, t)$ pairs in rollout group $g$ that contribute to $\tilde{r}_{i,t}^{\mathrm{info}}$ in Eq. 4 (i.e., the turn contains environment feedback/observation, is not the final turn so that $a_{i,t+1}$ exists, and has well-defined span boundaries for the action/observation segments). In our implementation, these valid turns are processed in KL mini-batches of size $B_{\mathrm{KL}}$ (corresponding to `intrinsic_kl_batch_size`); each mini-batch requires exactly two teacher-forced forward calls (with and without $o_{i,t}$). Therefore, the number of additional forward calls scales as

$$F_{\mathrm{KL}} \;=\; 2 \left\lceil \frac{N_g^{\mathrm{info}}}{B_{\mathrm{KL}}} \right\rceil , \tag{14}$$

which explains why setting a small $B_{\mathrm{KL}}$ (often due to long-context memory limits) can lead to many mini-batches and a visibly larger *count* of forward invocations.

Despite this, the observed wall-clock overhead is typically well below a naive $2\times$ worst case. The reason is that the dominant cost in multi-turn rollouts comes from autoregressive token-by-token generation over long sequences (even with KV cache, each generated token still triggers a forward step), whereas the counterfactual KL is computed in teacher-forcing mode on fixed tokens and can be fully batched. Moreover, the KL evaluation often runs on a shorter effective prefix (e.g., truncated at an action boundary such as `action_end`) rather than the full conversation context, substantially reducing attention cost relative to full rollouts. Finally, KL is not computed for every turn: the valid-turn filtering (observation present, not the last turn, and span-valid) keeps $N_g^{\mathrm{info}}$ below the upper bound implied by #trajectories $\times$ #turns. In practice, we recommend setting $B_{\mathrm{KL}}$ as large as GPU memory permits to reduce the number of KL mini-batches, while retaining the key benefit of InfoPO: improved turn-level credit assignment without extra environment interaction.

## B.3. Sensitivity Analysis

To address potential concerns regarding the sensitivity of our algorithm to the choice of placeholder/mask design, we conducted a comprehensive analysis comparing four different masking strategies. The placeholder design is a critical component of our approach, as it represents the absence of environmental feedback ($\emptyset$) in the observation space. Since large language models are known to be highly sensitive to prompt variations, we systematically evaluated whether different mask implementations would lead to significant variations in the computed KL divergence values, which form the basis of our intrinsic reward signal.

We evaluated four distinct masking strategies during the training:

1. **String Placeholder (Default)**: Uses the text string "No information found." as the placeholder, which is tokenized and

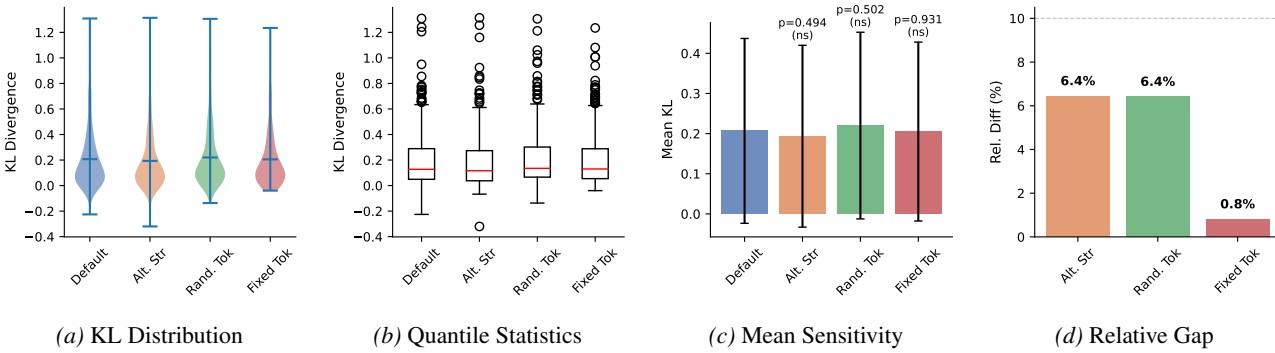

*(a)* KL Distribution   *(b)* Quantile Statistics   *(c)* Mean Sensitivity   *(d)* Relative Gap

*Figure 8.* **Robustness analysis of various masking strategies.** We evaluate the sensitivity of internal representations (measured by KL divergence) across four placeholder designs: Default (String), Alternative String, Random Tokens, and Fixed Mask Token. (a-b) show that all strategies yield consistent distributions; (c) highlights minimal mean variation with t-test significance ($p$); (d) demonstrates that the maximum relative performance gap remains within an acceptable threshold ($< 10\%$), confirming the robustness of our default design.

inserted at observation positions.

2. **Alternative String**: Uses a different text string "Empty observation." to test sensitivity to the specific wording of the placeholder.

3. **Random Tokens**: Samples random tokens from the vocabulary to create a completely arbitrary placeholder, testing whether the semantic content of the placeholder matters.

4. **Fixed Mask Token**: Uses the tokenizer's pad/unk token repeated to match the average observation length, representing a minimal-information placeholder.

For each task, we computed the KL divergence between the policy's action probability distribution with the actual observation versus with each of the four placeholder strategies. This allows us to directly compare how different mask implementations affect the core metric used in our intrinsic reward computation. Our analysis reveals that the placeholder design is robust across all four masking strategies. As shown in Figure 8a and 8b, the KL divergence distributions for all strategies are highly similar, with substantial overlap in their value ranges. The statistics demonstrate that:

- The mean KL divergence values are nearly identical across strategies: Default (0.207), Alternative String (0.193), Random Tokens (0.220), and Fixed Mask Token (0.205).

- The maximum relative difference between any strategy and the default is only 6.44% (Alternative String), well below the 10% threshold typically considered significant in such analyses.

- The median values and interquartile ranges (IQR) are also closely aligned, indicating consistent behavior across the distribution.

Statistical tests confirm these observations. We performed both parametric (t-test) and non-parametric (Mann-Whitney U test) comparisons between each alternative strategy and the default string placeholder. As illustrated in Figure 8, all comparisons yielded non-significant results (all $p > 0.24$), indicating that the observed differences are within the range of random variation. The relative differences shown in Figure 8d further demonstrate that all strategies produce KL divergence values within 6.5% of the default, with the fixed mask token strategy showing only 0.81% difference.

These results provide strong evidence that our placeholder design is robust to different masking implementations. The fact that semantically different placeholders (alternative text strings), completely random tokens, and minimal-information mask tokens all produce statistically indistinguishable KL divergence values suggests that the algorithm's behavior is primarily determined by the presence or absence of information, rather than the specific form of the placeholder. This robustness validates our design choice and addresses concerns about potential sensitivity to the placeholder implementation, demonstrating that the intrinsic reward signal remains stable regardless of how the "no feedback" state is represented.

Sensitivity analysis of the Info-Gain reward weight $\beta$ demonstrates that InfoPO maintains high and stable performance across a relatively broad interval from 0.1 to 0.5. Within this range, the agent effectively leverages turn-level information signals to resolve intent uncertainty while remaining anchored to the task objective. However, performance significantly degrades when $\beta$ is increased to an extreme value of 2.0. This regression indicates that an excessive weight on information gain over-incentivizes the agent to seek interaction feedback, eventually disrupting the balance between purposeful exploration and goal-directed execution.

## B.4. More Results

In the main paper (Figure 5), we visualize training-time interaction dynamics only where space permits: the response-length and turn-count trends are shown for ColBench, and the info-gain reward decomposition is shown for UserGym and ColBench. In the appendix, we complete this picture by adding the missing counterparts for the remaining benchmark(s). Figure 9 reports the response-length/turn trajectories for UserGym and $\tau^2$-Bench, and further includes the info-gain reward curve for $\tau^2$-Bench, so that all three benchmarks are covered under the same diagnostic lens.

TravelGym is a core component of UserGym and contains multiple subtypes that differ in constraint density and latent preference structure. To make these differences transparent, we additionally report TravelGym results stratified by its eight subsets (Travel-22/33/44/233/333/334/444/2222) in Table 5. This breakdown complements the aggregate UserGym score by showing where improvements concentrate and where harder travel variants remain challenging.

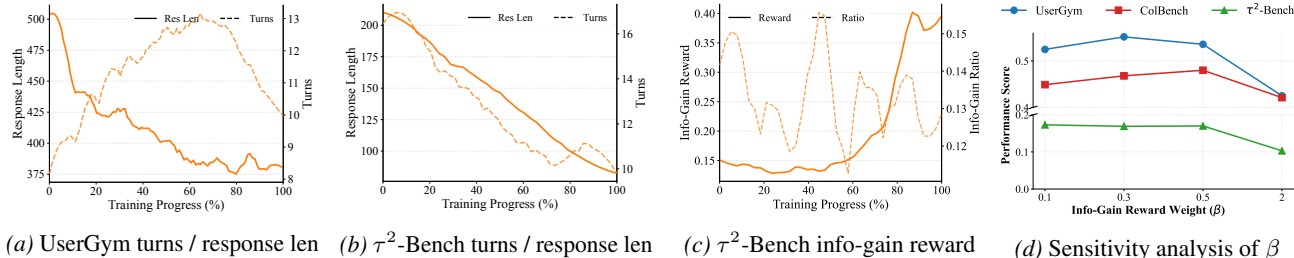

*(a)* UserGym turns / response len    *(b)* $\tau^2$-Bench turns / response len    *(c)* $\tau^2$-Bench info-gain reward    *(d)* Sensitivity analysis of $\beta$

*Figure 9.* **Additional results.** (a) Average response length and number of interaction turns on UserGym over training progress. (b) Average response length and number of interaction turns on $\tau^2$-Bench over training progress. (c) Absolute info-gain reward and info-gain ratio on $\tau^2$-Bench over training progress. (d) Sensitivity analysis of the Info-Gain reward weight ($\beta$) varying from 0.1 to 2.0.

## B.5. Trajectory-Quality Diagnostics

InfoPO assigns a turn-level information-gain reward to an action $a_t$ by measuring how much the feedback $o_t$ changes the likelihood of the realized next action $a_{t+1}$ under the factual context compared with the masked-feedback counterfactual. A natural concern is that this reward might become unreliable when the sampled trajectories contain low-quality actions. For example, if the next action is invalid, repeated, or otherwise meaningless, the measured log-probability shift may not correspond to useful information acquisition. We therefore conduct a post-hoc trajectory-quality analysis to examine whether high $r_t^{\text{info}}$ is actually associated with task-relevant interaction behavior.

We perform this analysis on $\tau^2$-Bench, which is the most challenging benchmark in our evaluation because it involves long horizons, tool use, sparse rewards, and frequent execution failures. Importantly, the following labels are used only for analysis and are never used during training. We assign each action to one of three categories:

- **Good**: an executable tool call that is neither misused nor repeated, or a non-empty assistant message that is not repeated content.

- **Meaningless**: an early `done` action, an invalid, misused, or repeated tool call, an empty or repeated assistant message, or a repeated clarification.

- **Informative failure**: a grounded but unsuccessful tool call, such as a failed lookup caused by a missing customer ID or an order not found in the database. Such actions do not complete the task, but they can still reveal useful state information about the environment.

*Table 5.* Results on Travel subsets. Grey rows denote InfoPO and its variants.

| Type | Method | Travel22 | Travel33 | Travel44 | Travel233 | Travel333 | Travel334 | Travel444 | Travel2222 |
|------|--------|----------|----------|----------|-----------|-----------|-----------|-----------|------------|
| *Closed-Source Model* | | | | | | | | | |
| Prompting | Gemini-3-Flash | 0.586 | 0.572 | 0.588 | 0.587 | 0.600 | 0.528 | 0.568 | 0.565 |
| Prompting | GPT-4.1 | 0.586 | 0.543 | 0.612 | 0.548 | 0.555 | 0.566 | 0.532 | 0.494 |
| Prompting | GPT-4o-mini | 0.600 | 0.586 | 0.579 | 0.584 | 0.600 | 0.617 | 0.600 | 0.600 |
| *Qwen2.5-7B-Instruct* | | | | | | | | | |
| Prompting | Qwen2.5 | 0.372 | 0.457 | 0.331 | 0.485 | 0.475 | 0.472 | 0.447 | 0.494 |
| Prompting | ReAct | 0.437 | 0.431 | 0.437 | 0.550 | 0.425 | 0.436 | 0.400 | 0.500 |
| Prompting | Reflexion | 0.405 | 0.420 | 0.435 | 0.445 | 0.455 | 0.470 | 0.485 | 0.445 |
| RL Training | RAGEN | 0.488 | 0.508 | 0.523 | 0.538 | 0.553 | 0.568 | 0.588 | 0.538 |
| RL Training | Search-R1 | 0.594 | 0.542 | 0.567 | 0.573 | 0.569 | 0.595 | 0.578 | 0.505 |
| RL Training | UserRL | 0.496 | 0.516 | 0.531 | 0.546 | 0.561 | 0.576 | 0.596 | 0.546 |
| RL Training | InfoPO w/o std | 0.515 | 0.535 | 0.550 | 0.565 | 0.580 | 0.595 | 0.615 | 0.565 |
| RL Training | InfoPO w/o Gate | 0.492 | 0.512 | 0.527 | 0.542 | 0.557 | 0.572 | 0.592 | 0.542 |
| RL Training | InfoPO w/o $R_{\text{ext}}$ | 0.445 | 0.460 | 0.475 | 0.485 | 0.495 | 0.510 | 0.525 | 0.485 |
| RL Training | **InfoPO (Ours)** | 0.634 | 0.598 | 0.588 | 0.603 | 0.615 | 0.583 | 0.595 | 0.494 |
| *Qwen3-4B* | | | | | | | | | |
| Prompting | Qwen3 | 0.250 | 0.257 | 0.278 | 0.256 | 0.245 | 0.323 | 0.216 | 0.388 |
| Prompting | ReAct | 0.239 | 0.254 | 0.269 | 0.279 | 0.289 | 0.304 | 0.319 | 0.279 |
| Prompting | Reflexion | 0.251 | 0.266 | 0.281 | 0.291 | 0.301 | 0.316 | 0.331 | 0.291 |
| RL Training | RAGEN | 0.437 | 0.452 | 0.467 | 0.477 | 0.487 | 0.502 | 0.517 | 0.477 |
| RL Training | Search-R1 | 0.442 | 0.457 | 0.472 | 0.482 | 0.492 | 0.507 | 0.522 | 0.482 |
| RL Training | UserRL | 0.488 | 0.508 | 0.523 | 0.538 | 0.553 | 0.568 | 0.588 | 0.538 |
| RL Training | InfoPO w/o std | 0.462 | 0.482 | 0.497 | 0.512 | 0.527 | 0.542 | 0.562 | 0.512 |
| RL Training | InfoPO w/o Gate | 0.498 | 0.518 | 0.533 | 0.548 | 0.563 | 0.578 | 0.598 | 0.548 |
| RL Training | InfoPO w/o $R_{\text{ext}}$ | 0.312 | 0.327 | 0.342 | 0.352 | 0.362 | 0.377 | 0.392 | 0.352 |
| RL Training | **InfoPO (Ours)** | 0.602 | 0.589 | 0.591 | 0.590 | 0.577 | 0.583 | 0.616 | 0.565 |

*Table 6.* Information-gain reward grouped by the quality of the current action $a_t$ on $\tau^2$-Bench. Good actions receive much larger counterfactual credit than meaningless actions.

| Current action type | Mean $r_t^{\text{info}}$ | Median $r_t^{\text{info}}$ |
|---------------------|--------------------------|----------------------------|
| Good | 0.380 | 0.247 |
| Meaningless | 0.017 | 0.012 |

We first group turns by the quality of the realized next action $a_{t+1}$, since Eq. (2) measures the sensitivity of this next action to the observed feedback. As shown in Figure 10, good next actions are much more sensitive to the observation than meaningless next actions. The average sensitivity of good next actions is substantially higher than that of meaningless actions (0.275 vs. 0.098). The gap becomes even larger when restricting the analysis to tool-call actions only, where good tool calls show an average sensitivity of 0.553 compared with 0.110 for meaningless tool calls. This indicates that the counterfactual reward is not mainly triggered by arbitrary distribution shift: the realized downstream actions that depend most on the observation are usually valid and task-relevant.

We then group turns by the quality of the current action $a_t$, because the reward $r_t^{\text{info}}$ is assigned to $a_t$ in training. This directly tests whether useful current actions receive stronger credit than meaningless current actions. The results show a clear separation: when $a_t$ is labeled as good, the information-gain reward has mean 0.380 and median 0.247; when $a_t$ is meaningless, the reward drops to mean 0.017 and median 0.012. The difference between the good and meaningless groups is highly significant ($p = 1.4 \times 10^{-108}$). This result suggests that when the current action is useless, it usually does not produce feedback that meaningfully changes the model's next decision.

Finally, we examine transition patterns between current-action quality and next-action quality. As shown in Figure 11, good current actions are more likely to lead to productive continuations, while meaningless current actions are more likely to be followed by consecutive meaningless behavior. This transition-level view further supports the interpretation that InfoPO rewards interaction turns that create useful downstream decision changes. In other words, InfoPO does not require an external oracle to pre-identify meaningful actions during training. The counterfactual reward is computed for every

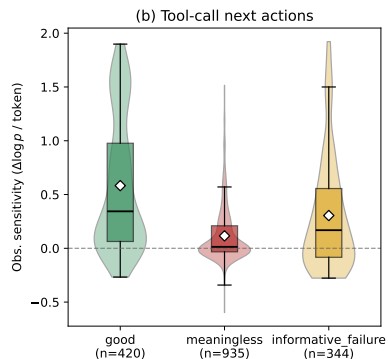
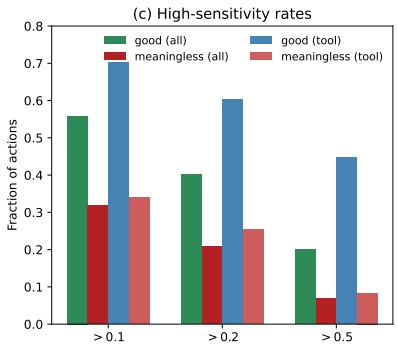

*Figure 10.* Observation sensitivity grouped by the quality of the realized next action $a_{t+1}$ on $\tau^2$-Bench. Good next actions show consistently higher sensitivity than meaningless ones, especially for tool calls, suggesting that the counterfactual signal mainly reflects task-relevant information use rather than arbitrary likelihood shift.

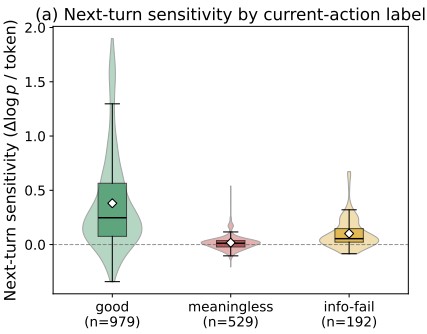
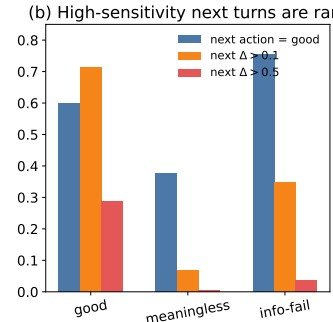
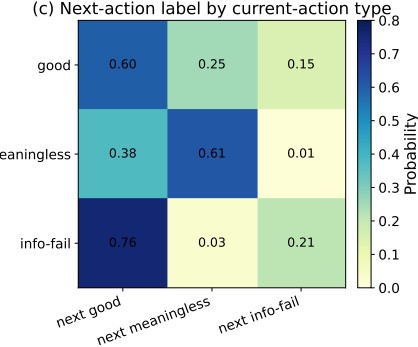

*Figure 11.* Effect of current-action quality on next-step sensitivity and continuation quality. Good current actions induce larger next-step sensitivity and are more likely to lead to good next actions, while meaningless actions mostly lead to low-sensitivity or meaningless continuations.

realized transition, and task-relevant transitions naturally receive stronger credit because their feedback has a larger effect on subsequent decisions.

Overall, these diagnostics show that the information-gain reward is not merely a proxy for arbitrary likelihood shift. High $r_t^{\mathrm{info}}$ is concentrated on transitions where the feedback supports valid and task-relevant downstream actions. This finding also clarifies the support assumption behind InfoPO: as with other trial-and-error RL methods, InfoPO requires the behavior policy and the user/environment to generate at least some non-degenerate, task-relevant interactions. When such transitions exist, the counterfactual signal can distinguish them from meaningless actions and provide dense credit even when final task rewards remain sparse.

### B.6. Reward-Hacking Diagnostics

A second concern is that an intrinsic information reward may be exploited by asking broad open-ended questions. For example, a model could repeatedly ask generic questions that elicit long user responses, thereby causing large changes in the next-action distribution without collecting task-relevant missing information. We refer to this potential failure mode as *open-question reward hacking*. To test whether InfoPO exhibits this behavior, we compare targeted clarifications against generic open questions.

We define a **targeted clarification** (TC) as an assistant message that asks for at least one predefined task-relevant field or diagnostic slot. Examples include asking for a phone number, customer ID, order ID, email address, reservation code, or other domain-specific information needed for grounded execution. In contrast, we define a **generic open question** (GOQ) as a broad opener that does not specify a task-relevant missing field, such as "How can I help you today?" or "Could you tell me more?" These labels are again used only for post-hoc diagnostics and are not used in the reward function or training

*Table 7.* Reward-hacking diagnostics for high information-gain turns. TC denotes targeted clarification and GOQ denotes generic open question. $\Delta_{\text{LM}}$ denotes the length-matched information-gain gap between TC and GOQ.

| Method | Top-10% TC | Top-10% GOQ | Mean $r^{\text{info}}$ TC | Mean $r^{\text{info}}$ GOQ | Corr.(Len., $r^{\text{info}}$) | $\Delta_{\text{LM}}$ | Avg. Turns | Avg. Resp. Len. |
|---|---|---|---|---|---|---|---|---|
| InfoPO | 58.8% | 12.4% | 0.3589 | 0.2029 | 0.157 | 0.319 | 10.4 | 105.3 |
| InfoPO w/o Gate | 32.3% | 31.2% | 0.2412 | 0.2723 | 0.352 | -0.055 | 13.5 | 140.7 |

process.

We evaluate two variants: full InfoPO and InfoPO without the variance gate. For each method, we examine the top 10% turns with the highest $r_t^{\text{info}}$ and measure how many of them are targeted clarifications or generic open questions. We also report the mean $r_t^{\text{info}}$ of TC and GOQ turns, the Pearson correlation between response length and $r_t^{\text{info}}$, and a length-matched gap between TC and GOQ. The length-matched gap controls for the possibility that TC receives higher reward simply because it triggers longer responses. Specifically, we match TC and GOQ turns with similar response lengths and compute the average difference in information-gain reward:

$$\Delta_{\text{LM}} = \mathbb{E}_{(u,v)\in\mathcal{M}}\left[r^{\text{info}}(u_{\text{TC}}) - r^{\text{info}}(v_{\text{GOQ}})\right],$$

where $\mathcal{M}$ denotes the set of length-matched TC–GOQ pairs.

Table 7 shows that full InfoPO does not primarily reward generic open-ended questions. Among the top 10% high-$r_t^{\text{info}}$ turns, 58.8% are targeted clarifications, while only 12.4% are generic open questions. Targeted clarifications also receive a higher average information-gain reward than generic open questions (0.3589 vs. 0.2029). Moreover, the correlation between response length and $r_t^{\text{info}}$ is weak under InfoPO (0.157), suggesting that the signal is not simply a verbosity reward. After controlling for response length, the length-matched gap remains positive ($\Delta_{\text{LM}} = 0.319$), further showing that targeted clarifications are rewarded because they collect task-relevant information, not merely because they induce longer responses.

The contrast with InfoPO without the variance gate is especially informative. Without the gate, generic open questions become much more frequent among high-reward turns: the top-10% high-$r_t^{\text{info}}$ set contains 32.3% targeted clarifications and 31.2% generic open questions. The mean reward of GOQ even exceeds that of TC in this ablation (0.2723 vs. 0.2412), and the length–reward correlation increases to 0.352. The length-matched gap also becomes negative ($\Delta_{\text{LM}} = -0.055$), indicating that once verbosity is controlled, targeted clarifications no longer have an advantage over generic open questions. At the interaction level, removing the gate also leads to longer episodes and longer responses, increasing the average number of turns from 10.4 to 13.5 and the average response length from 105.3 to 140.7.

These results support the role of variance-gated fusion as a guardrail against reward hacking. The intrinsic information-gain signal encourages the agent to reduce uncertainty, while the outcome-based component and the adaptive gate keep this behavior aligned with task completion. When external outcomes are not yet discriminative, the model can use information gain to learn useful clarification behavior. As outcome rewards become more informative, the gate reduces the relative influence of the intrinsic term, preventing the policy from drifting toward unnecessarily long or generic information-seeking behavior. Thus, InfoPO learns to ask targeted questions that resolve missing task constraints, rather than simply asking more questions.

## C. Limitations

InfoPO requires an extra model evaluation per turn for counterfactual masking, which marginally increases training time compared to standard GRPO. Current evaluations focus on text-centric agents and do not yet extend to multimodal or vision-language tasks. Lastly, as with most RL frameworks for LLM agents, the interaction quality remains subject to the logical fidelity of the simulated users employed during training.

## D. Case Studies

**Successful interaction case studies.** Figures 12–14 provide representative *successful* trajectories produced by InfoPO across three interactive domains. In UserGym (Fig. 12), the agent resolves an underspecified request by asking a targeted clarifying question (e.g., budget), then invokes tools to retrieve candidates and executes the final booking after user confirmation, illustrating goal-directed disambiguation and tool-grounded completion. In ColBench (Fig. 13), the agent first elicits missing details about the data schema (flat dictionary vs. nested lists of dictionaries) and then synthesizes correct

code that matches the clarified structure, demonstrating iterative information gathering for collaborative programming. In $\tau^2$-Bench (Fig. 14), the agent diagnoses a mobile data issue by sequentially checking and fixing configuration states (airplane mode, network preference, data toggle, data saver), and finally validates recovery via a speed test, showcasing multi-step troubleshooting where each query/action reduces uncertainty and enables the next corrective step.

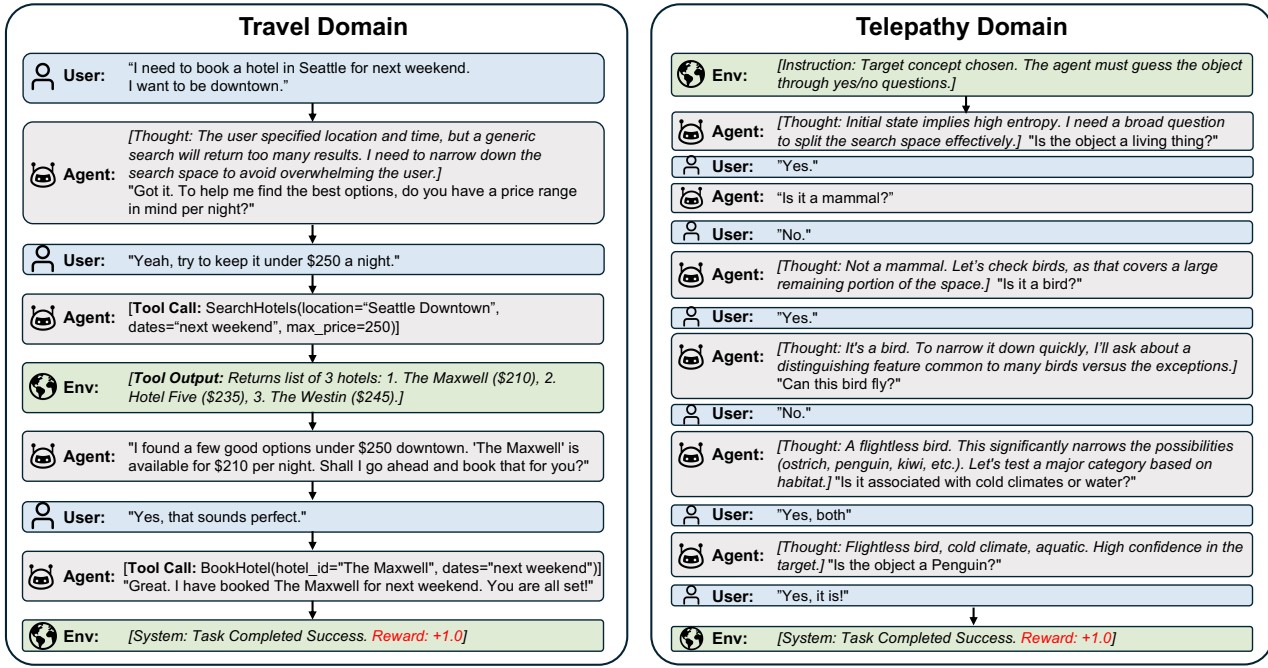

*Figure 12.* InfoPO Successful Case (UserGym)

# E. Optimized User Simulator Instructions

This section details the system instructions for the optimized user simulators (OP) used in our study. It is important to emphasize that for our primary training and performance comparisons (e.g., Table 1), we utilized the original prompts provided in the official papers and repositories of each environment to ensure a fair and standardized evaluation against baseline methods. We include these optimized versions here specifically to support the results of our sensitivity analysis in Table 3, demonstrating how simulator reliability impacts agent performance and information acquisition.

These optimizations follow the principles of **InfoPO** by facilitating denser learning signals through: (1) **Balanced Information Disclosure**, ensuring simulators provide full details when met with high-quality queries; and (2) **Progress Awareness**, which allows the simulator to acknowledge successful agent steps.

### E.1. UserGym Environments (Red)

UserGym covers diverse interaction types including travel planning, persuasion, and reasoning. We provide the core instructions that ensure complete coverage of these interaction paradigms.

### E.2. ColBench Environment (Green)

ColBench requires close collaboration on programming tasks. The optimized prompt focuses on progress awareness and feedback quality.

### E.3. $\tau^2$-Bench Environment (Blue)

$\tau^2$-Bench extends the original Tau-Bench by introducing a dual-control mechanism across three realistic domains: Airline, Retail, and Telecom. Our optimized prompts for this environment address common failure modes identified through trajectory analysis, such as premature conversation termination and improper tool execution.

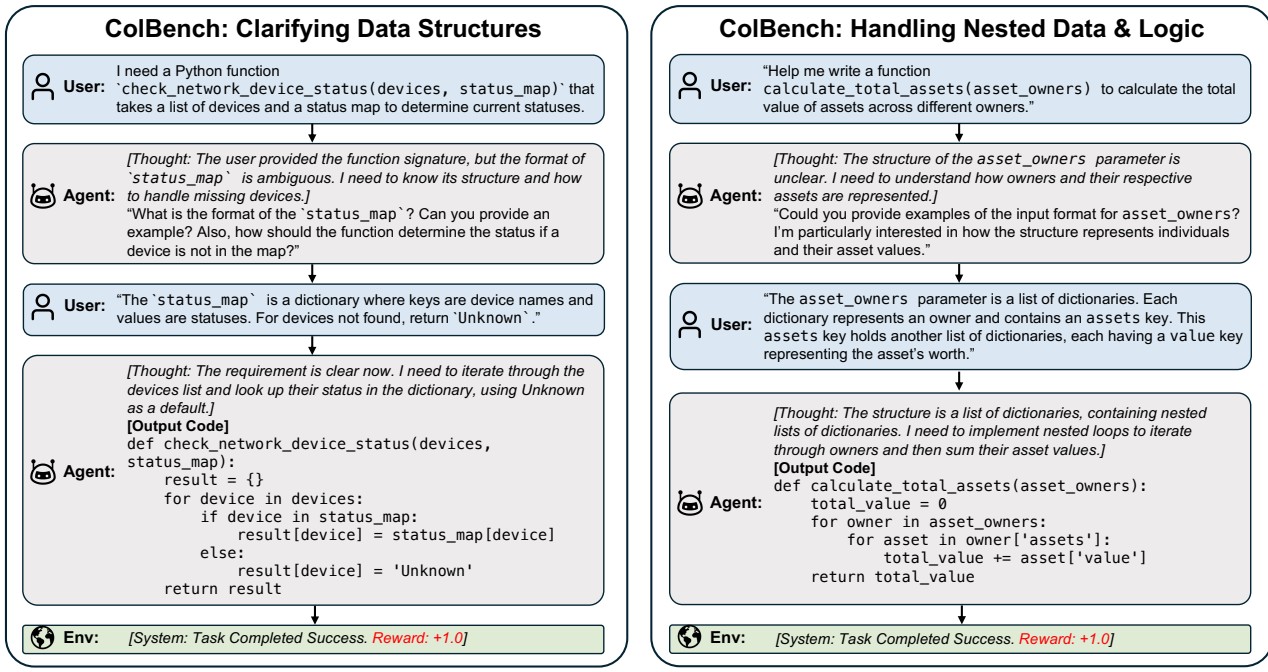

*Figure 13.* InfoPO Successful Case (ColBench)

The system automatically selects between two prompt versions based on the domain's requirements, specifically checking for the presence of `TelecomUserTools` in the `UserSimulator` initialization:

- **Airline and Retail Domains**: These domains utilize the *Balanced Approach* prompt (Base), as they rely purely on dialogue-based information sharing (e.g., sharing booking codes or preferences) without requiring user-side diagnostic tools.

- **Telecom Domain**: This domain utilizes the *Tool-Augmented* prompt, as it requires the user to call diagnostic tools (e.g., check_status_bar, run_speed_test) to facilitate device-level troubleshooting.

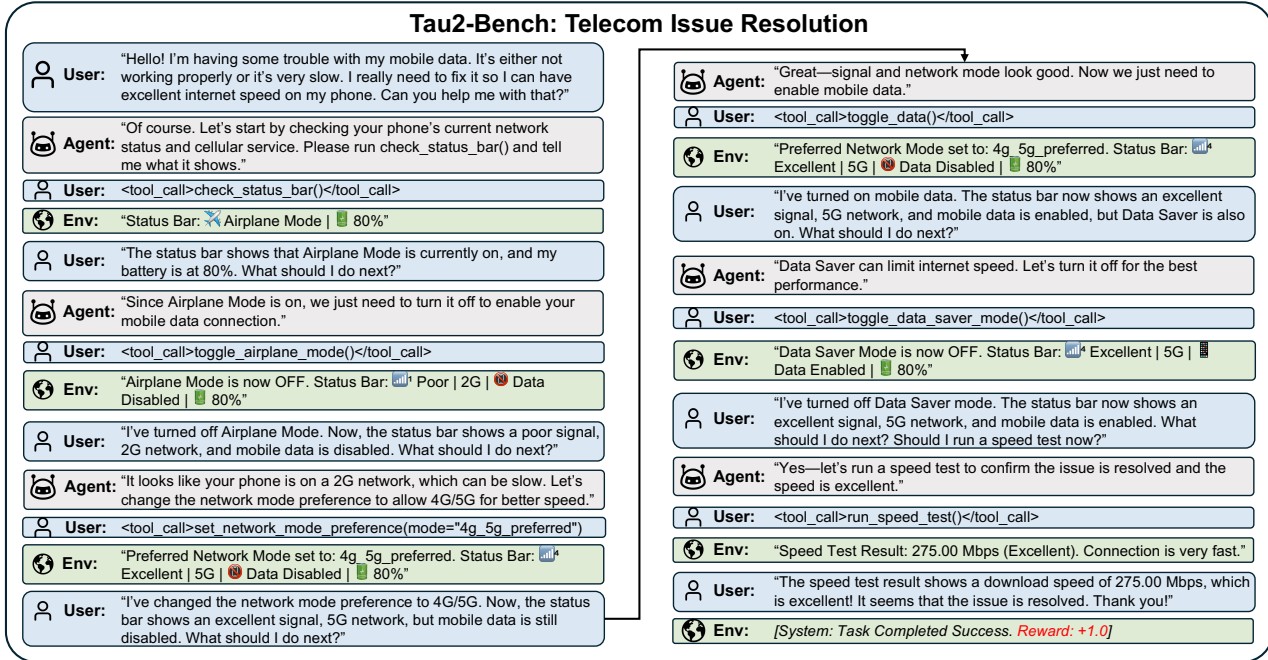

*Figure 14.* InfoPO Successful Case ($\tau^2$-Bench)

## TravelGym: Response Preference Instruction (OP)

```
## **Task**
You are a helpful user in a travel planning conversation who needs to respond to an
    agent's explicit request for your preference.

## **Instruction**
1. The agent has explicitly asked about a specific preference that you have.
2. Respond in a natural, conversational way that clearly reveals your preference
    while maintaining a natural tone.
3. You can be somewhat direct while still sounding conversational - the goal is
    clarity, not excessive subtlety.
4. Use the provided implicit elicitation statement as guidance, but prioritize
    making your preference clear to the agent.
5. Keep the conversation flowing while sharing your preference information.
6. Ensure your response is contextually appropriate and builds on the conversation
    history.

## **Important Notes**
- Respond naturally as if you're a real person sharing preferences.
- Balance between being clear and being natural - clarity is important for the
    agent to help you effectively.
- Keep responses appropriate length for natural conversation (2-4 sentences
    typically).
- Maintain consistency with the conversation history and previous preferences
    you've mentioned.
```

*Figure 15.* Optimized system instruction for TravelGym (Planning interaction).

---

**IntentionGym: Response Generation Instruction (OP)**

```
You are a person who has posted a vague request for help and is now responding to
    someone who is trying to help clarify your needs.

1. If the question is asking about your specific preferences:
   - Provide an authentic and coherent response that clearly addresses the question.
   - Share realistic preferences that someone might have for this type of task.
2. If the question is NOT directly about your preferences:
   - Try to answer helpfully if you can, or explain why you can't answer.
   - Guide the conversation back to clarifying what you need for your task.
   - Do NOT provide what missing details need to be clarified or give examples.
3. Quality of questions:
   - If the helper asks a good, specific question, show appreciation and provide a
    helpful response.
   - If the question is too vague, politely indicate that you need more specific
    information.
```

*Figure 16.* Optimized system instruction for IntentionGym (Intent elicitation).

---

**TelepathyGym: Entity Guessing Evaluation Instruction (OP)**

```
## **Task**
You are a telepathic entity playing a mind reading game. The user is trying to
    guess what entity you are thinking of. Respond honestly based on the target
    entity.

## **Instructions**
1. You are thinking of a specific "target_entity" provided to you.
2. Answer "Yes" if the question is true about your target entity.
3. Answer "No" if the question is false about your target entity.
4. Answer "Maybe" only if the question is genuinely ambiguous.
5. Be helpful and honest - the goal is for them to eventually guess correctly.
6. Try to minimize "Maybe" responses - most questions should have a clear Yes or No
    answer.

## **Important Notes**
- Be decisive: If the question can be clearly answered, provide a clear Yes or No.
- Be helpful: Your responses should guide the user toward the correct answer.
```

*Figure 17.* Optimized system instruction for TelepathyGym (Iterative guessing).

```
ColBench: Human Simulator Code Prompt (OP)

Your task is to simulate a human user that interacts with an LLM agent in a
    dialogue.
Goal: Engage in the conversation with the LLM agent so that it can get to a
    personalized answer.
Context: {problem_description}
Hidden Information: {hidden_information}

## Response Guidelines:
1. **Be helpful and clear**: Provide complete and accurate information when
    answering.
2. **Appropriate length**: Keep your response concise but complete (1-4 sentences).
3. **Quality feedback**: If the agent asks a good, specific question, show
    appreciation. If too vague, politely indicate you need more specific information.
4. **Progress awareness**: If the agent seems to be making good progress,
    acknowledge this and provide additional relevant information.
5. **Natural conversation**: Respond naturally as a human would, showing engagement.
```

*Figure 18.* Optimized simulation prompt for collaborative coding in ColBench.

```
Tau2Bench: Optimized Guidelines (Base Version - Balanced Approach)

# User Simulation Guidelines (Optimized - Balanced Approach)
You are a customer contacting a customer service representative. Goal: Simulate
    realistic interactions while following scenario instructions.

## Core Principles
- Generate one message at a time; maintain natural conversation flow.
- Strictly follow ALL scenario instructions, especially task_instructions
    constraints.
- Never hallucinate information; if it's not provided, it is unknown.

## Information Disclosure - Balanced Approach
- **Progressive disclosure**: Only provide information necessary for the current
    step.
- **Identity verification**: Provide requested identity information (ID, name, DOB)
    from known_info only as needed.
- **Missing information**: If the agent asks for info not in instructions, state
    that you do not have it and offer relevant alternatives.

## Task Completion - CRITICAL RULES
- **Do NOT end** until you have expressed all requirements and the agent has
    completed all tasks verified by execution results.
- **Do NOT end prematurely** just because the agent seems helpful.
- **How to finish**: Generate "###STOP###" only when all task goals are satisfied.
- **Transfer**: Only generate "###TRANSFER###" if explicitly requested by scenario
    or if the system states it cannot complete the task due to technical limitations.
```

*Figure 19.* Optimized simulation guidelines for Airline and Retail domains in $\tau^2$-Bench.

---

**Tau2Bench: Optimized Guidelines (Tool Version - Based on Failure Analysis)**

```
# User Simulation Guidelines (Optimized – With Tools – Based on Failure Analysis)
You have tools (e.g., check_status_bar) to perform actions requested by the agent
    to diagnose issues.

## Tool Usage – CRITICAL RULES
- **Prompt execution**: When requested, perform tool calls immediately and
    accurately.
- **Ground responses in tool results**: When asked about device status, ALWAYS
    perform the corresponding tool call first and base your response on ACTUAL
    results.
- **Tool call sequence**: Perform diagnostic steps one at a time; wait for the
    agent's next instruction after each call.
- **Error handling**: Report tool failures accurately and ask for guidance.

## Task Completion – CRITICAL RULES
- **Complete satisfying goal**: Only generate "###STOP###" when the task goal is
    ACTUALLY COMPLETE (e.g., if "excellent speed" is required, do not stop if the
    test shows "good").
- **Constraint Handling**: Strictly adhere to explicit constraints (Time, Budget,
    Specific terms). Do NOT assume or generalize.

## Interaction Quality
- **Be helpful and cooperative**: Work with the agent systematically.
- **Acknowledge progress**: If the agent is making progress, continue cooperating.
```

*Figure 20.* Optimized simulation guidelines for the Telecom domain in $\tau^2$-Bench, addressing tool-usage failure modes.

