# OpenReview forum: "InfoPO: Information-Driven Policy Optimization for User-Centric Agents"
_ICML.cc/2026/Conference — ICML 2026 regular_

### Official Review · Reviewer_B8ue · 2026-03-02

**Soundness:** 4
**Presentation:** 4
**Significance:** 3
**Originality:** 2
**Overall Recommendation:** 4
**Confidence:** 4

**Summary:**

This paper addresses the challenge of credit assignment in multi-turn LLM agents, specifically when user requests are underspecified and require interaction to resolve. The authors introduce InfoPO (Information-Driven Policy Optimization), a framework that treats multi-turn interaction as a process of active uncertainty reduction. By defining a turn-level counterfactual information-gain reward, the model is trained to identify actions that elicit the most useful feedback from a user or environment. This intrinsic signal is combined with task outcomes via an adaptive variance-gated fusion to ensure the agent remains goal-oriented while learning to be more inquisitive.

**Compliance With Llm Reviewing Policy:**

Affirmed.

**Key Questions For Authors:**

1. All evaluations used GPT-4-based simulators. How do the authors expect the "information-gain" signal to hold up against the noise and irrationality of actual human users compared to the simulators used in the study?
2. In reinforcement learning, agents can sometimes get distracted by information that is high-gain but irrelevant to the goal. How does the variance-gated fusion specifically prevent the agent from becoming "over-inquisitive" about interesting but task-irrelevant user details?
3. This may be related to previous question: The choice of intrinsic weight (\beta) and gate temperature (T) varies significantly across benchmarks. Can the authors provide some intuition or a "rule of thumb" for setting these values in new environments?
4. The current work is text-centric. Given that GUI-based or multimodal agents often face even greater uncertainty, have the authors anticipated any hurdles in adapting the counterfactual masking approach to visual observations or complex action spaces?

**Limitations:**

yes

**Strengths And Weaknesses:**

**Strengths**

The introduction of turn-level information gain as a reward is a thoughtful way to address the sparse reward problem in long-horizon agent tasks. By providing denser feedback at each step of interaction, the method encourages more purposeful exploration rather than delayed credit assignment. The paper also theoretically shows that cumulative information gain is a necessary resource for task success and connects the reward design to conditional mutual information. The authors recognize that maximizing information gain can conflict with task rewards and introduce adaptive variance-gated fusion mechanism to address.

**Weaknesses**

A practical limitation is the need for an additional model forward pass per turn for counterfactual masking, which increases computational overhead. This may reduce the method’s attractiveness in resource-constrained or real-time settings. Performance also appears sensitive to the intrinsic weight and gate temperature. Selecting these parameters currently depends on domain-specific knowledge of reward variance, which may limit robustness across environments. Finally, the overall interaction quality depends heavily on the fidelity of the LLM-simulated users used during training. If the simulator exhibits inconsistencies or unrealistic behavior, the agent may internalize suboptimal interaction strategies.

---

> ### Author Rebuttal · Authors · 2026-03-31
>
> We sincerely appreciate the reviewer’s constructive feedback. We address your concerns in detail below.
> > W1: Computational overhead
>
> We agree that InfoPO introduces additional computation. However, in practice, the added computational overhead is usually much smaller than a complete extra rollout per sample, because the main training cost still comes from rollout generation, while our counterfactual computation is done **on fixed tokens and can be efficiently parallelized in batch**. We also notice that similar trade-offs also appear in recent self distillation methods like SDPO (Hübotter et al., 2026), which uses additional teacher/student-style forward passes and KL-based distillation to obtain more fine-grained learning signals.
>
> > W2 & Q3: The value of intrinsic weight and gate temperature
>
> We thank the reviewer for the careful observation on InfoPO. We have already explained the choice of the two parameters in **Appendix B.1**, and **Fig. 9(d)** provides a sensitivity analysis for $\beta$. More specifically, $\beta$ controls the overall strength of the info-gain signal, while $T$ controls how quickly it should give way to the extrinsic signal once the task outcomes within a rollout group become discriminative. Based on our experiments, a simple practical rule is that if the extrinsic reward of a task is sparse, but becomes highly discriminative once it appears like $\tau^2$-Bench, then a lower $T$ is usually more suitable. $\beta$ is relatively stable as long as it stays below 1.
>
> > W3 & Q1: LLM-simulated users
>
> First, we think that for the user-centric tasks we study, some level of user rationality and capability is inherently required, regardless of whether the user is a real human or an LLM simulator. To check this more directly, we ran a simple experiment on $\tau^2$-Bench where we replaced the user simulator with **a much smaller model, Qwen2.5-0.5B**. We found that even strong closed-source models such as Gemini-3-Flash and GPT-4.1 drop to **almost zero** under this setting.
>
> At the same time, for the GPT-4o-mini level simulator used in our main training setup, its **low cost makes large-scale rollout collection practical**, and it also means that its feedback **naturally contains noise**. To further quantify this, and to analyze how our method behaves under different user-feedback quality levels, we selected two representative environments, TravelGym from UserGym and Telecom from $\tau^2$-Bench, and define three categories of user feedback:
> - `informative`: the user provides task-grounded, actionable preference, state, or constraint information that can directly affect the next action;
> - `weak`: the user is still cooperative, but the feedback is partial, underspecified, unavailable, repeated, or otherwise lower-value;
> - `poor`: the current user reply does not provide a usable signal for the next action.
>
> We then analyze trajectories collected under our main setup, where the agent is Qwen-2.5-7B and the user simulator is GPT-4o-mini. We report both the proportion of these three feedback types and the effect of feedback quality at turn t on the turn-level info-gain reward. The results are as follows:
>
> |Environment|I %|W %|P %|$\Delta_I$|$\Delta_W$|$\Delta_P$|$p(I,W)$|$p(I,P)$|$p(W,P)$|
> |-|-|-|-|-|-|-|-|-|-|
> |TravelGym|59.9%|23.5%|16.6%|0.307|0.142|0.019|3.84e-04|7.04e-12|6.58e-05|
> |Telecom|77.8%|15.1%|7.1%|0.377|0.359|0.155|8.34e-02|2.82e-04|1.05e-07|
>
> Here, I/W/P denote informative / weak / poor user feedbacks, and $\Delta$ is the **raw turn-level counterfactual information signal, i.e., $r_t^{info}$.**
>
> These results show that GPT-4o-mini **does produce noisy feedback**, and InfoPO **does not simply reward any user utterance**. For example, in TravelGym, the average next-step sensitivity drops from 0.307 for informative feedback to 0.142 for weak feedback, and further to only 0.019 for poor feedback. All three pairwise differences are statistically significant. So when the user provides useful feedback that directly changes the next action, the information-gain reward becomes larger.
>
> > Q2: Reward hacking
>
> We have already addressed the similar reward-hacking concern in detail in our response to **`Reviewer ipvf`**. Due to the rebuttal word limit, we kindly refer you to that rebuttal for the full analysis.
>
> > Q4: Visual tasks or complex action spaces
>
> We agree this is an important direction, but it is outside the scope of the current paper. Actually, the core idea of InfoPO only requires comparing the same next action before and after masking part of the observation, so the framework could extend to multimodal agents with observation-level masking over visual or UI state. The main challenges are how to control the higher counterfactual evaluation cost for long visual token sequences, and how to obtain a stable turn-level signal when the action space becomes more complex. We will clarify this scope limitation and discuss these extension challenges more explicitly in the revision.

---

> > ### Author Rebuttal · Reviewer_B8ue · 2026-04-01
> >
> > My concerns are addressed.

---

> > > ### Author Response · Authors · 2026-04-03
> > >
> > > Dear Reviewer B8ue,
> > >
> > > Thank you very much for your thoughtful review and your encouraging feedback on our work. We are grateful that our rebuttal was able to address your concerns.
> > >
> > > We sincerely appreciate the time and effort you devoted to reviewing our submission. Your constructive comments were very helpful to us and have contributed to improving the paper.
> > >
> > > Thank you again for your support and consideration.
> > >
> > > Best regards,
> > >
> > > Authors

---

### Official Review · Reviewer_t7B1 · 2026-03-08

**Soundness:** 3
**Presentation:** 2
**Significance:** 3
**Originality:** 3
**Overall Recommendation:** 4
**Confidence:** 5

**Summary:**

While in real-world user requests to LLM agents are often underspecified, it is important to enable LLM agents to proactively acquire information from the user and make downstream decisions, such that the agents would be able to capture the correct and true intention of the user request. In this paper propose InfoPO, a policy optimization method that claims to a) capture the information gain per step and b) jointly consider the final outcome reward to guide the policy towards the correct direction in the RL training. This method is tested in several circumstances to prove its validity.

**Compliance With Llm Reviewing Policy:**

Affirmed.

**Final Justification:**

The rebuttal has addressed my concerns, and I hope my comments could be helpful to improve the readability and technical solidity of the work.

**Key Questions For Authors:**

Please see the weakness part above. In addition, I am very curious if the authors tested larger models (like qwen2.5/3-32b)? Would the improvements be more significant or negligible? And how is that \beta in (5) chosen? What is the impact of the choice of value?

I may consider adjusting my rating if my concerns are abundantly addressed.

**Limitations:**

yes

**Strengths And Weaknesses:**

## Strengths

* The paper tackles an important and practical issue: real-world user requests to LLM agents are often underspecified, so agents should proactively acquire missing information and make downstream decisions to better capture the user’s true intent.

* The framing of proactive interaction as a game of information acquisition is intuitive. In many realistic settings, the primary proactive behavior is “asking,” and the key challenge is how to measure the quality of the interaction, i.e., the quality of the questions.

* Although the dec-POMDP modeling itself is not particularly novel, the paper presents an interesting reward/advantage design. The turn-level counterfactual information gain, computed as the average log-probability shift over the next action tokens, is clearly conveyed (Fig. 1 helps readers quickly grasp the core idea).

* The method is evaluated under several circumstances, which helps demonstrate its validity.

* Experiments are comprehensive.

## Weaknesses

* Some parts of the method appear to rely on an implicit assumption about the existence of “meaningful trajectories” during interaction with users or simulators. In particular, to compute the turn-level counterfactual information gain, it seems necessary to assume the next “action” of the agent is "meaningful" (such as 'cancel(abc123)'). If the action is "not meaningful", then the placeholder information (the "meaningless") may change neglibible probability distribution? If so, one needs to first have the next meaningful action identified. How would that be done?

* Relatedly, for a group G, if some action is “not good” (i.e., it does not trigger a useful reaction from the user/simulator), it is unclear how r_t^{info} behaves. The value might be non-trivial; could this lead training in a wrong direction when trajectories are problematic?

* The outcome and info-gain advantages are fairly standard; the fusion component is the more interesting part. The adaptive gain g(.) seems to handle edge cases (e.g., when sigma^ext_g is nearly zero), but it is unclear how robust it is when the sampled trajectories themselves are problematic (as raised above).

* Overall, the design seems to assume that for all tasks the agent (LLM) can react in “good” ways with reasonable probability. If this assumption is required, it would help for the authors to state it explicitly and provide a justification or discussion of failure modes when it does not hold.

* The two theorems are straightforward and appear to be variants of Fano’s inequality. It is not clear what role they play in the paper beyond adding theoretical depth? The authors may want to clarify their purpose and how they inform the method or its guarantees.

---

> ### Author Rebuttal · Authors · 2026-03-31
>
> We sincerely appreciate the reviewer’s constructive feedback. We address your concerns in detail below.
>
> > W1-4: The assumption about the existence of “meaningful trajectories”
>
> First, our method itself does not require an explicit distinction between meaningful and meaningless actions. However, like any trial-and-error RL method, it **cannot work if the policy is completely degenerate and fails to roll out any useful data at all**. To address this concern more directly, we conducted the following additional analysis on $\tau^2$-Bench because it is the hardest and most failure-prone of the three environments. It has long horizons, tool use, and sparse rewards. We assign each action to one of three categories:
>
> - `good` means an executable tool call that is not a misuse or repeated call, or a non-empty assistant message that is not repeated content.
> - `meaningless` means an early done, an invalid, misused, or repeated tool call, an empty or repeated message, or a repeated clarification.
> - `informative` failure means a grounded but unsuccessful tool call, such as customer/order not found, which do not succeed but still reflect state-related exploration.
>
> First, for the concern about **(1) the next action should be meaningful**, we collected offline trajectories generated by the base Qwen2.5-7B model and computed $\log p(a_{t+1}\mid h_t, o_t) - \log p(a_{t+1}\mid h_t, \mathrm{mask}(o_t))$. We find that next actions labeled as `good` are much more sensitive to the observation, and significantly more so than `meaningless` actions. The mean sensitivity is **0.275** for good versus **0.098** for meaningless actions, with $p = 1.2 \times 10^{-20}$. Restricting to tool calls only, the gap becomes even larger, at 0.553 versus 0.110.
>
> **Detailed results are shown in this figure:** https://ibb.co/cSj53pRP
>
> Second, for **(2) the effect of the current action $a_t$ on $r_t^{\mathrm{info}}$**, we group turns by the quality of $a_t$, and then measure the sensitivity of the next action to $o_t$, which is exactly what defines $r_t^{\mathrm{info}}$. We find that if $a_t$ is `good`, the $r_t^{\mathrm{info}}$ is high, with **mean 0.380 and median 0.247**. If $a_t$ is meaningless, it drops to **mean 0.017 and median 0.012**, with $p = 1.4 \times 10^{-108}$. Therefore, when the current action does not trigger a useful reaction, it usually does not receive a strong $r_t^{\mathrm{info}}$. We believe this further supports the design of the InfoPO reward.
>
> **Detailed results are shown here**: https://ibb.co/fYGNdfBv
>
> These two results give a direct and intuitive picture of why $r^{\mathrm{info}}$ is useful. In addition, Table 2 in the main paper already shows that $\sigma_g^{\mathrm{ext}}$ is not always zero in practice, so the fusion module does have real opportunities to act. Overall, we agree that InfoPO needs model to generate at least some reasonable actions, rather than complete nonsense. But we view this as a general requirement of RL-based methods. Under this condition, which we believe is realistic for current powerful LLMs, the info-gain reward does provide a meaningful learning signal. We will make this assumption and these results more explicit in the revision.
>
> > W5: Theorems
>
> Thank you for this suggestion. We agree that Theorems 1 and 2 should not be interpreted as formal guarantees for InfoPO's convergence or for the entire RL process. They provide a principled motivation for our reward design. In particular, they explain why reducing uncertainty about the hidden task state can improve downstream decision quality. Our method operationalizes this idea through a turn-level information-gain reward, while the variance-gated fusion balances this auxiliary signal with task-level feedback.
>
> To make this role clearer, we also examined the empirical relationship between intrinsic information gain and extrinsic task outcomes. Across the three environments, we observe consistently **positive Pearson correlations of 0.822 on ColBench, 0.624 on tau2, and 0.643 on UserGym**. This suggests that the intrinsic signal is meaningfully aligned with task success in practice, which is also supported in the above rebuttal. We will revise the paper to clarify this positioning more explicitly.
>
> > Q1: Larger models like qwen2.5/3-32b
>
> We agree that experiments at a larger scale would be interesting. However, under our resource setup in **Section 5.1 (4×80GB GPUs)** and the high memory demand of long sequences in multi-turn tasks (for example, in $\tau^2$-Bench, the average sequence length is **over 7k tokens**, and some trajectories even exceed **20k tokens**.), **7B-scale** models are already the largest models we could run stably and obtain results from within a reasonable time.
>
> > Q2: Choice of $\beta$
>
> We have already provided **a sensitivity analysis for $\beta$ in Fig. 9d**. The main finding is that performance is relatively stable when $\beta$ is in the range of 0.1 to 0.5, while a clear drop appears when it is further increased to 2.

---

> > ### Author Rebuttal · Reviewer_t7B1 · 2026-04-02
> >
> > Many thanks for your detailed clarifications, and most of my concerns have been fully resolved. As a practitioner in production-level RL, I am always happy to see solid work in the field, especially those that focus on solving practical problems.

---

> > > ### Author Response · Authors · 2026-04-03
> > >
> > > Dear Reviewer t7B1,
> > >
> > > Thank you very much for your thoughtful review and your encouraging feedback on our work. We are grateful that our rebuttal was able to address your concerns.
> > >
> > > We sincerely appreciate the time and effort you devoted to reviewing our submission. Your constructive comments were very helpful to us and have contributed to improving the paper.
> > >
> > > Thank you again for your support and consideration.
> > >
> > > Best regards,
> > >
> > > Authors

---

### Official Review · Reviewer_ipvf · 2026-03-13

**Soundness:** 3
**Presentation:** 3
**Significance:** 3
**Originality:** 3
**Overall Recommendation:** 4
**Confidence:** 4

**Summary:**

This paper presents InfoPO, which uses turn-level counterfactual information gain for better credit assignment problem in multi-turn GRPO based methods.

**Compliance With Llm Reviewing Policy:**

Affirmed.

**Key Questions For Authors:**

- I wonder if the authors have examined or analyzed the data on those high info-gain rewards during training? it would be good to verify if they are of reward hacking trajectories or actual newly inspired generations.

- In fig 8, all KL distributions are similar among different strategies, I wonder if authors have looked into if this are random OOD inputs or actual information gain?

**Limitations:**

Yes

**Strengths And Weaknesses:**

### Strengths

- This paper is generally well written and easy to follow that tackles a known problem. By introducing the Turn-level Info-gain reward, this allows that if an agent successfully responds with good outpus to a user, it is rewarded for that specific turn, even if the final task isn't perfectly completed.

- The design of masked-feedback counterfactual on turn level, plus the use of Adaptive Variance-Gated Fusion is novel and straightforward for solving the issue of reward hacking and providing accurate round evaluation.

- The experiment design is done along with comprehensive baselines and ablation. The results show clear performance improvements.

### Weakness

- Since one of the core idea of this paper is on counter factual information by masking, feeding LLM with random inputs with placeholders in the middle of training can obviously push the model output and training outside of previous distribution, however this increasd KL divergence does not necessarily mean something useful is learnt/found.

- Since the info-gain reward is gathered based on the shift from the models policy, one obvious reward hacking is that to generate some random open question that will most likely bring long/different respose from the user simulator.

---

> ### Author Rebuttal · Authors · 2026-03-31
>
> We sincerely appreciate the reviewer’s constructive feedback. We address your concerns in detail below.
>
> > W1 & Q2: Concern about counterfactual information gain and its KL distributions
>
> First, in Figure 8, the four placeholders are not all random inputs. Two of them are **coherent and meaningful strings**, such as “no information found”, while the other two are a random-token placeholder and a fixed mask token. The fact that InfoPO is not sensitive to these differences already suggests that the main factor is the information carried by the true observation, rather than the exact form of the placeholder itself.
>
> Second, the design of the info-gain reward is one of the main features that distinguishes InfoPO from GRPO. The main-paper experiments already show that this design is effective and useful for completing the tasks.
>
> Finally, to further address this concern, we conducted an additional analysis on $\tau^2$-Bench. We assign each action to one of three categories:
> - `good` means an executable tool call that is not a misuse or repeated call, or a non-empty assistant message that is not repeated content.
> - `meaningless` means an early done, an invalid, misused, or repeated tool call, an empty or repeated message, or a repeated clarification.
> - `informative failure` means a grounded but unsuccessful tool call, such as customer/order not found. Such actions do not succeed, but they still reflect state-related exploration.
>
> We group turns by the quality of $a_t$, and then measure the counterfactual info-gain reward defined in Eq. (2). We found that if $a_t$ is `good`, the reward is high, with **mean 0.380 and median 0.247**. If $a_t$ is `meaningless`, it drops to **mean 0.017 and median 0.012**, with a highly significant difference from the `good` group ($\mathbf{p = 1.4 \times 10^{-108}}$). It shows that when the current action is useless, it usually does not receive a strong $r_t^{\mathrm{info}}$.
> We also examined the **transition pattern** from the current action type to the next one. We find that `good` actions are much more likely to lead to productive continuation, whereas `meaningless` actions are more likely to trigger consecutive `meaningless` behavior. We therefore believe this analysis provides direct evidence that the counterfactual reward captures task-relevant information gain, rather than merely reflecting arbitrary distribution shift.
>
> ***Detailed results are shown in this figure: https://ibb.co/fYGNdfBv***
>
> > W2 & Q1: Reward hacking
>
> As shown above, we already showed that high info-gain rewards are associated with more useful actions. However, we agree that, in principle, an intrinsic information signal could be exploited because the model might keep asking open questions. This is exactly why we use **variance-gated fusion** instead of adding the intrinsic term with a fixed weight.
>
> We further explicitly define `targeted clarification (TC)` as an assistant message that asks for at least one predefined task-relevant field or diagnostic slot, such as phone number, customer id, order id, or email, and `generic open question (GOQ)` as a question that matches a broad opener, such as “How can I help you today?”. The results show that, under InfoPO, actions with **high $r^{info}$ are much more concentrated in `targeted clarifications`**.
> We also compute the overall correlation between observation length and $r^{info}$ and find that this correlation is weak. Further, we perform a near length-matched comparison between targeted clarifications and generic open questions, showsing that the advantage of targeted clarification is not driven by length alone.
> The detailed results are as follows:
>
> | Method | Top-10% $r^{info}$: TC | Top-10% $r^{info}$: GOQ | Mean $r^{info}$ of TC | Mean $r^{info}$ of GOQ | Corr(response length, $r^{info}$) | Length-matched Delta $r^{info}$ (TC - GOQ) | Avg turns | Avg response length |
> |----|--|--|---|--|--|---|-|-|
> | InfoPO | 58.80%  | 12.40% | 0.3589  | 0.2029  | 0.157| 0.319  | 10.4 | 105.3   |
> | InfoPO w/o Gate  | 32.30%    | 31.20%  | 0.2412   | 0.2723  | 0.352| -0.055 | 13.5 | 140.7 |
>
> These results directly address the reward-hacking concern. Under InfoPO, high $r^{\mathrm{info}}$ is much more concentrated on targeted clarifications than on generic open questions (58.8% vs. 12.4% in the top-10% reward turns). The weak correlation with response length (0.157) and the positive length-matched gap (+0.319) further show that the signal is closely tied to collecting task-relevant missing information. We also see a clear contrast with InfoPO w/o Gate, where generic open questions become much more common, interactions become longer, responses become longer, and the length-matched advantage of targeted clarification disappears.
>
> Overall, the high-reward turns in InfoPO are mostly useful, task-grounded actions that help move the interaction toward task completion.

---

> > ### Author Rebuttal · Reviewer_ipvf · 2026-04-02
> >
> > I thank the authors for their responses. They have resolved my concerns, I'll remain my original score.

---

> > > ### Author Response · Authors · 2026-04-03
> > >
> > > Dear Reviewer ipvf,
> > >
> > > Thank you very much for your thoughtful review and your encouraging feedback on our work. We are grateful that our rebuttal was able to address your concerns.
> > >
> > > We sincerely appreciate the time and effort you devoted to reviewing our submission. Your constructive comments were very helpful to us and have contributed to improving the paper.
> > >
> > > Thank you again for your support and consideration.
> > >
> > > Best regards,
> > >
> > > Authors

---

### Official Review · Reviewer_u8x9 · 2026-03-16

**Soundness:** 3
**Presentation:** 3
**Significance:** 3
**Originality:** 3
**Overall Recommendation:** 4
**Confidence:** 2

**Summary:**

This paper demonstrates that concepts from information theory can be ported over to shed light on numerous aspects of multi-turn agent interaction, including both credit assignment in sparse-reward settings and the theoretical underpinnings of active uncertainty reduction. From a more technical perspective, this paper shows empirically that replacing outcome-only advantage estimation with a turn-level counterfactual information gain reward can result in improved task performance and training stability across diverse interactive benchmarks.

**Compliance With Llm Reviewing Policy:**

Affirmed.

**Final Justification:**

My questions were fully addressed.

**Key Questions For Authors:**

1.Have you considered applying attention suppression combined with KL divergence to potentially provide a more direct measure of the causal impact of an observation on the policy’s next-action distribution, instead of relying on textual placeholders for counterfactual estimation?
2.Why was this particular sigmoid-based gate chosen? Were other gating strategies or annealing schedules considered?
3.Have you evaluated alternatives (e.g., KL between next-action distributions), and how do they compare in stability and compute?

**Limitations:**

yes

**Strengths And Weaknesses:**

# Strengths
1.Introduces a scalable, task-agnostic turn-level information-gain reward with variance-gated fusion, grounded in information theory, to address long-horizon credit assignment in sparse-reward multi-turn tasks.
2.Evaluated across multiple user-centric benchmarks and non-interactive tasks, with ablations, stability diagnostics, and robustness checks demonstrating effectiveness and transferability.
3.Algorithm components, fusion mechanism, and training setup are described in detail.
# Weaknesses
There may be some minor issues.
1.Counterfactual estimation via textual placeholders may introduce semantic confounding.
2.Variance-gated fusion uses a sigmoid with fixed output 0.5 at zero extrinsic variance; the choice of functional form and calibration lacks justification or sensitivity analysis.

---

> ### Author Rebuttal · Authors · 2026-03-31
>
> We sincerely appreciate the reviewer’s constructive feedback. We address your concerns in detail below.
>
> > W1 & Q1: Counterfactual estimation
>
> We agree that a placeholder-based counterfactual can raise the concern that the measured shift may depend on the specific wording of the placeholder. So **we have already examined this issue in Appendix B.3 (Fig. 8)** through a placeholder-sensitivity analysis. Specifically, we compare four masking strategies: the default string placeholder, an alternative string placeholder, random-token placeholders, and fixed mask-token placeholders. Across all four variants, the resulting information-gain statistics remain highly consistent: the largest relative difference is only 6.44%, and all pairwise tests are **non-significant (all p > 0.24)**, suggesting that the signal is driven primarily by information removal, rather than by the specific semantics of the placeholder. We will make this point more explicit in the revision.
>
> Regarding the alternative you mentioned, internal interventions such as attention suppression are indeed interesting. However, our goal is to define the counterfactual at the interaction level, that is, how the policy’s next-action preference changes when the observation is removed from the dialogue history.  Text masking implements this intervention directly while keeping the model and inference pipeline unchanged.  Attention suppression instead modifies internal computation, which makes it a different causal intervention and introduces architecture-specific choices.  We therefore use textual counterfactuals as the more model-agnostic and benchmark-agnostic design.
>
> > W2 & Q2: The design of gate function
>
> We chose the sigmoid gate mainly because it provides a **smooth, bounded, and monotonic** way to combine info-gain and outcome reward. We think that gating based on the within-group variance of the external reward provides more detailed reward shaping than a global anneal schedule based on training steps. The reviewer is  correct that the sigmoid gate gives $g(0)=0.5$ when $\sigma^{ext}=0$. However, this midpoint should not be interpreted as fixing the intrinsic contribution at 0.5 in practice, since the final intrinsic term is further scaled by $\beta$ in Eq. (5). We also provide **a sensitivity analysis for $\beta$ in Appendix Fig. 9(d).** To further address this concern, we compare four variants on UserGym: sigmoid (ours), a linear gate, a linear annealing schedule, and w/o Gate. The results show that the group-adaptive gates are consistently stronger than both step-based annealing and removing the gate entirely.
>
> | Fusion strategy| UserGym avg.|
> |--|--|
> |Sigmoid gate (ours)| **0.552**|
> |Linear gate| 0.537|
> |Linear anneal| 0.521|
> |w/o Gate| 0.495|
>
> > Q3: Alternatives about KL between next-action distributions
>
> We thank the reviewer for raising the meaningful alternatives. Our current estimate is a **teacher-forced log-probability shift** computed on the realized next-action tokens. We chose this method mainly for two reasons.
>
> First, in our long-horizon setting, the exact sequence-level KL between next-action distributions is intractable, since it would require marginalizing over all possible autoregressive continuations. A more practical alternative would be a token-level full-vocabulary KL under teacher forcing. However, this would still be more expensive, because it requires recording and computing the full vocabulary distribution at every next-action position under both the factual and counterfactual contexts.
>
> Second, our goal is specifically to measure **whether the observation changes the policy’s confidence for the downstream action** actually taken in the sampled trajectory. In this sense, a realized-token log-probability shift is more closely aligned with trajectory-level credit assignment. By contrast, a full-vocabulary KL also captures probability mass shifts over tokens that are never selected.
>
> We will discuss this point in more detail in the revision.

---

> > ### Author Rebuttal · Reviewer_u8x9 · 2026-04-03
> >
> > I thank the authors for their responses. They have resolved my concerns, and I will maintain my original score.

---

> > > ### Author Response · Authors · 2026-04-03
> > >
> > > Dear Reviewer u8x9,
> > >
> > > Thank you very much for your thoughtful review and your encouraging feedback on our work. We are grateful that our rebuttal was able to address your concerns.
> > >
> > > We sincerely appreciate the time and effort you devoted to reviewing our submission. Your constructive comments were very helpful to us and have contributed to improving the paper.
> > >
> > > Thank you again for your support and consideration.
> > >
> > > Best regards,
> > >
> > > Authors

---

### Decision · Program_Chairs · 2026-04-30

**Decision:**

Accept (regular)

**Comment:**

The paper provides a novel algorithm that uses turn-based counterfactual information gain for credit assignment problem in multi-turn GRPO.
One point that the reviewers pointed out is the need of meaningful trajectories. Although I see the point of the authors, one of the RL core hardness is the design of policies that can collect meaningful data (that it is not solved in the paper). I suggest to add a discussion about this in the paper.